# Rethinking Graph Transformers as Graph Signal Denoisers: The Role of Block-Diagonal Priors

Jiaming Zhuo[1]  Ziyi Ma[1]  Kun Fu[1]  Di Jin[2]  Chuan Wang[3]  Zhen Wang[4]
Xiaochun Cao[5]  Huazhu Fu[6]  Liang Yang[1][*]

## Abstract

By synergizing graph topology with the global expressive power of attention, Graph Transformers (GTs) have emerged as a powerful architecture for node classification. Existing GTs mainly focus on diverse topology injection paradigms, which fundamentally construct different propagation operators. However, a unified theoretical understanding of what constitutes a desirable propagation operator remains largely unexplored. To bridge this gap, this paper rethinks GTs from a graph signal denoising perspective, revealing that a block-diagonal structure is a desirable structural prior for graph denoising. To efficiently instantiate this prior, this paper introduces BDFormer, which imposes spectral block regularization on the affinity of a compact set of latent anchors. Furthermore, by introducing hard assignments between nodes and anchors, BDFormer establishes *sparse*, *structured* propagation that suppresses noisy cross-class propagation in linear time. Meanwhile, the learned global affinity guides the pruning of inter-class edges in the graph topology, enabling both global and local propagation to jointly adhere to the target block-diagonal structure. Extensive experiments on benchmark datasets demonstrate the superiority and scalability of BDFormer.

[1]Hebei Province Key Laboratory of Big Data Calculation, School of Artificial Intelligence, Hebei University of Technology, Tianjin, China. [2]Tianjin Key Laboratory of Cognitive Computing and Application, College of Intelligence and Computing, Tianjin University, Tianjin, China. [3]School of Computer Science and Technology, Beijing JiaoTong University, Beijing, China. [4]School of Artificial Intelligence, OPtics and ElectroNics (iOPEN), School of Cybersecurity, Northwestern Polytechnical University, Xi'an, China. [5]School of Cyber Science and Technology, Shenzhen Campus of Sun Yat-sen University, Shenzhen, China. [6]Institute of High Performance Computing, Agency for Science, Technology and Research, Singapore. Correspondence to: Liang Yang <yangliang@vip.qq.com>.

*Proceedings of the 43rd International Conference on Machine Learning*, Seoul, South Korea. PMLR 306, 2026. Copyright 2026 by the author(s).

## 1. Introduction

Node classification, aimed at predicting the labels of individual nodes by learning expressive representations from node attributes and graph topology, is an essential task across diverse domains, such as social networks (Tabassum et al., 2018) and citation networks (Zhao et al., 2022; He et al., 2024; Zhuo et al., 2024; Zhao et al., 2024; Yang et al., 2025b; Huang et al., 2025; Yang et al., 2025a; Huang et al., 2026; Wang et al., 2026). Benefiting from their powerful modeling capabilities, Graph Neural Networks (GNNs) have become a standard architecture for this task (Kipf & Welling, 2017; Velickovic et al., 2017; Zhou et al., 2025b;a). Most GNNs follow the Message-Passing (MP) paradigm (Gilmer et al., 2017), where node representations are iteratively updated by aggregating features from local neighborhoods (Guan et al., 2025a;b; Liang et al., 2026). Although this paradigm provides GNNs with local inductive biases, modeling long-range dependencies (Dwivedi et al., 2022) across distant nodes typically requires deep stacking, which may result in over-smoothing and over-squashing (Giraldo et al., 2023).

Recently, Transformers (Vaswani et al., 2017) have shown great promise across multiple domains such as natural language processing and computer vision (Gillioz et al., 2020; Han et al., 2022). Building on this architecture, particularly the Self-Attention (SA) mechanism that models global, all-pairs interactions, Graph Transformers (GTs) have been developed and have emerged as a powerful architecture for node classification (Min et al., 2022; Zhuo et al., 2025a; Chen et al., 2024; 2025). The success of GTs is largely attributed to coupling SA with structural inductive biases from the graph topology. By enabling interactions beyond immediate neighborhoods, GTs provide a compelling alternative to these neighborhood-restricted GNNs.

Depending on the paradigm of topology injection, existing GTs can be broadly grouped into two categories: (1) *Score-level* topology injection, which modulates the attention scores either implicitly via positional encodings (Kreuzer et al., 2021; Rampásek et al., 2022; Chen et al., 2023) or explicitly via architectural mechanisms, such as structural biases (Ying et al., 2021) and masking mechanisms (Shirzad et al., 2023; Liu et al., 2024); and (2) *Representation-level*

topology injection, which derives topology-aware node representations from a GNN module and fuses them with the SA output (Wu et al., 2023; Deng et al., 2024). To fully comprehend these diverse architectures, recent research efforts have attempted to unify them. For instance, UGC-Former (Zhuo et al., 2025b) identifies cross-aggregation as the shared mechanism, revealing that GTs essentially model the interaction between graph topology and node attributes. Meanwhile, Xing et al. (2025) unify GTs under a hierarchical masking perspective, mapping architectures to specific attention ranges. Despite their distinct realizations, these topology injection paradigms fundamentally operate by constructing different propagation operators. However, existing unification efforts remain primarily mechanism-oriented, lacking a principled understanding for the question:

*What constitutes a desirable propagation operator for Graph Transformers?*

To answer this question, this paper rethinks GTs from a Graph Signal Denoising (GSD) perspective, where the propagation operator acts as the core denoising operator governing how node signals diffuse. Under this interpretation, the denoising quality fundamentally depends on the structural quality of the propagation operator. However, existing GTs typically lack explicit structural constraints on this operator. Consequently, topology-defined operators inherently absorb topological noise (*e.g.*, heterophilous edges) from the raw graph, while attention-defined operators suffer from dense all-pair mixing that introduces severe propagation noise. Both unconstrained dynamics inevitably lead to spurious cross-class propagation, degrading the denoising quality.

Drawing upon the homophily principle (McPherson et al., 2001), the ideal propagation operator naturally manifests as a block-diagonal matrix (Lu et al., 2018), where propagation is primarily concentrated within the same class, as illustrated in Fig. 1(a). Theoretically, such a structure serves as a favorable GSD prior by suppressing noisy cross-class propagation while preserving intra-class information aggregation. However, directly enforcing a block-diagonal constraint on the $n \times n$ node-level affinity matrix is computationally prohibitive. To efficiently realize this block-diagonal prior, this paper proposes BDFormer. Specifically, it introduces a compact set of latent anchors and imposes spectral block regularization on their affinity structure, encouraging the emergence of block-wise propagation patterns. Based on these regularized anchors, discrete hard assignments establish sparse node-anchor interactions that suppress spurious cross-class propagation in the global channel with linear complexity. Meanwhile, the learned anchor-induced affinity further refines the graph topology by pruning inter-class edges, enabling both global and local propagation to synergistically conform to the target block-diagonal structure.

The major contributions of this paper can be outlined as

- We rethink GTs under a graph signal denoising perspective, establishing that a block-diagonal structure acts as an ideal structural prior for feature propagation.

- We propose BDFormer to efficiently instantiate the block-diagonal prior via spectral anchor regularization and discrete hard assignments, which jointly enable a sparse, structured feature propagation.

- Extensive evaluations across diverse homophilic, heterophilic, and large-scale graphs demonstrate that BD-Former achieves state-of-the-art performance while scaling linearly with respect to the graph size.

## 2. Preliminaries

This section first introduces the employed notations. It then reviews the basic Transformer and categorizes existing Graph Transformers by their topology injection paradigms.

### 2.1. Notations

This paper considers an undirected attributed graph $\mathcal{G} = (\mathcal{V}, \mathcal{E})$, where $\mathcal{V}$ and $\mathcal{E}$ denote the node set and edge set, respectively. $\mathcal{V}$ contains $n$ node instances $\{(\mathbf{x}_v, \mathbf{y}_v)\}$, where $\mathbf{x}_v \in \mathbb{R}^{1 \times f}$ and $\mathbf{y}_v \in \mathbb{R}^{1 \times c}$ represent the $f$-dimensional node attributes (signals) and $c$-dimensional labels, respectively. Typically, the graph topology is represented by the adjacency matrix $\mathbf{A} \in \mathbb{R}^{n \times n}$, with $a_{i,j} = 1$ only if the edge $(v_i, v_j) \in \mathcal{E}$ and $a_{i,j} = 0$ otherwise. Therefore, the graph can be reformulated as $\mathcal{G} = (\mathbf{A}, \mathbf{X})$. $\mathbf{D} = \mathrm{diag}(\mathbf{A1})$ and $\mathbf{L} = \mathbf{I} - \mathbf{D}^{-\frac{1}{2}} \mathbf{A} \mathbf{D}^{-\frac{1}{2}}$ stand for the diagonal degree matrix and the normalized Laplacian matrix, respectively. For semi-supervised tasks, the provided labels are represented by $\mathbf{Y}_L \in \mathbb{R}^{n_L \times c}$, where $n_L$ is the number of labeled nodes.

### 2.2. Transformers

While originally developed for sequence modeling (Vaswani et al., 2017), the Transformer architecture has been generalized as a universal backbone for processing structured data, *e.g.*, graphs (Min et al., 2022). Fundamentally, it leverages the *Attention Module* to model all-pair interactions. Besides, *Residual Connections* (He et al., 2016) and *Normalization Layers* (Xiong et al., 2020) are commonly employed.

**Attention Module.** The attention module generally models interactions between a query set and a source set. Formally, given query representations $\mathbf{H}_Q \in \mathbb{R}^{n_q \times d}$ and source representations $\mathbf{H}_S \in \mathbb{R}^{n_s \times d}$, the module projects them into the Query ($\mathbf{Q}$), Key ($\mathbf{K}$), and Value ($\mathbf{V}$) matrices, that is,

$$\mathbf{Q} = \mathbf{H}_Q \mathbf{W}_Q, \quad \mathbf{K} = \mathbf{H}_S \mathbf{W}_K, \quad \mathbf{V} = \mathbf{H}_S \mathbf{W}_V, \quad (1)$$

where $\mathbf{W}_Q, \mathbf{W}_K, \mathbf{W}_V$ denote learnable parameter matrices. The scaled dot-product attention function $\mathrm{Attn}(\mathbf{Q}, \mathbf{K}, \mathbf{V})$

computes the normalized attention weights $\mathbf{S}$ and performs a weighted aggregation of the Value vectors, defined as

$$\text{Attn}(\mathbf{Q}, \mathbf{K}, \mathbf{V}) = \mathbf{S}\mathbf{V} = \text{softmax}\left(\hat{\mathbf{S}}\right) \mathbf{V}$$
$$= \text{softmax}\left(\frac{\mathbf{Q}\mathbf{K}^\top}{\sqrt{d}}\right) \mathbf{V}. \quad (2)$$

This formulation unifies both *Self-attention* ($\mathbf{H}_Q = \mathbf{H}_S$) and *Cross-attention* ($\mathbf{H}_Q \neq \mathbf{H}_S$). In practice, multi-head attention is adopted to capture diverse dependency patterns.

### 2.3. Graph Transformers (GTs)

GTs generalize the Transformer architecture to graphs by incorporating topological inductive biases. Depending on the paradigm of topology injection, existing GTs can be categorized into two distinct paradigms: Score-level injection and representation-level injection.

**Score-level Topology Injection.** This type of GTs injects graph topology into the attention module by adjusting the pre-softmax attention scores. Although the implementation forms vary, these designs share a common objective: to construct a *structure-aware affinity graph* where edge weights are jointly determined by feature similarity and topological proximity. Formally, the unnormalized score $\hat{s}_{i,j}$ (*i.e.*, the $(i,j)$-th entry of $\hat{\mathbf{S}}$) can be formulated as

$$\hat{s}_{i,j} = (\mathbf{q}_i, \mathbf{k}_j) + g(\phi_{i,j}) + m_{i,j} \quad (3)$$

where $\kappa(\mathbf{q}_i, \mathbf{k}_j)$ denotes the content-based similarity, which implicitly captures topological priors when positional encodings (*e.g.*, (Kreuzer et al., 2021; Rampásek et al., 2022; Chen et al., 2023)) are augmented onto the node attributes. $g(\phi_{i,j})$ encodes explicit pairwise structural information via additive soft biases (*e.g.*, shortest path distances in Graphormer (Ying et al., 2021)). $m_{i,j}$ abstracts structural masking, encompassing additive log-masks (*i.e.*, $m_{i,j} \in \{0, -\infty\}$) for strict sparsity (*e.g.*, Exphormer (Shirzad et al., 2023)) and multiplicative decay variants (*e.g.*, Gradformer (Liu et al., 2024)).

**Representation-level Topology Injection.** This paradigm injects topological information directly into node representations via an auxiliary GNN module. Specifically, it fuses global representations derived from the attention module with local representations from the GNN, formulated as

$$\mathbf{H}^+ = \mathcal{F}\left(\mathbf{H}_{\text{Attn}}, \text{GNN}(\mathbf{A}, \mathbf{H})\right), \quad (4)$$

where $\mathcal{F}(\cdot, \cdot)$ denotes a fusion operator (*e.g.*, weighted combination). Representative GTs include SGFormer (Wu et al., 2023) and M$^3$Dphormer (Xing et al., 2025).

## 3. Methodology

This section first unifies existing Graph Transformers (GTs) under a Message Passing (MP) formulation, which is further

interpreted through a Graph Signal Denoising (GSD) perspective. Then, it presents BDFormer to efficiently impose a block-diagonal prior on the propagation operator, followed by a comprehensive analysis of this architecture.

### 3.1. GTs as Graph Signal Denoisers

Despite their architectural distinctions, both score-level and representation-level designs can be formulated as a unified MP step on a weighted affinity graph. Importantly, this formulation admits a GSD-inspired interpretation, revealing the implicit optimization objective governing these models.

**A Unified Message Passing Formulation.** Given the initial node features $\mathbf{H}^{(0)} \in \mathbb{R}^{n \times d}$, and omitting feature projections, normalization, and MLPs for clarity, a broad class of MP updates at the $l$-th layer is formulated as

$$\mathbf{H}^{(l)} = (1 - \tau)\mathbf{H}^{(0)} + \tau\mathbf{P}\mathbf{H}^{(l-1)}, \quad (5)$$

where $0 \leq \tau \leq 1$ is a tradeoff scalar and $\mathbf{P} \in \mathbb{R}^{n \times n}$ denotes the propagation operator (corresponding to the underlying affinity structure) that determines the propagation dynamics.

**Score-level *vs.* Representation-level Topology Injection.** The distinction between existing GT designs lies primarily in how the propagation operator $\mathbf{P}$ is specified.

*Remark* 1 (Implicitly learned propagation in score-level topology injection paradigms). The propagation operator is instantiated directly as the normalized attention matrix, that is, $\mathbf{P} = \mathbf{S}$ (in Eq. 2), obtained by applying row-wise softmax to the unnormalized scores $\hat{s}_{i,j}$. This enables global, content-adaptive feature propagation.

*Remark* 2 (Explicit and topology-constrained propagation in representation-level topology injection paradigms). Conversely, this paradigm defines $\mathbf{P}$ based on the explicit graph topology. Here, the operator is fixed to a structural matrix (*e.g.*, the normalized adjacency $\tilde{\mathbf{A}}$ or a pre-computed diffusion kernel (Klicpera et al., 2019)), thereby restricting the feature propagation to the structural neighborhood.

**Optimization Perspective.** The unified message passing formulation in Eq. 5 is not merely a heuristic update; rather, it is interpreted as a graph signal denoising (GSD) process, where $\mathbf{P}$ functions as a diffusion operator regulating the smoothness of node signals (Shuman et al., 2013).

**Proposition 1** (First-Order Approximation of GSD)**.** *Let* $\mathbf{P} \in \mathbb{R}^{n \times n}$ *be a non-negative row-stochastic matrix defining the affinity graph, which is typically asymmetric due to mechanisms like softmax. Consider the GSD objective* $\min_{\mathbf{Z} \in \mathbb{R}^{n \times d}} \mathcal{L}(\mathbf{Z})$*, where the loss function is defined as:*

$$\mathcal{L}(\mathbf{Z}) = \frac{1}{2}\|\mathbf{Z} - \mathbf{H}^{(0)}\|_F^2 + \frac{\lambda}{2}\text{tr}\left(\mathbf{Z}^\top(\mathbf{I} - \mathbf{P})\mathbf{Z}\right). \quad (6)$$

*Minimizing this objective via a single-step gradient descent update, which preserves the directed information flow of* $\mathbf{P}$*,*

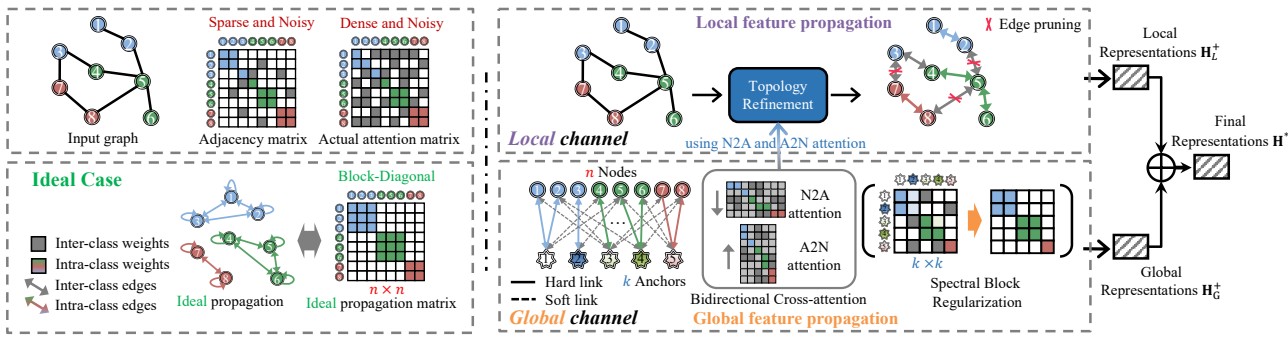

(a) Design Motivation: Noisy vs. Ideal.  (b) Architecture of BDFormer.

*Figure 1.* Overview of BDFormer. (a) Motivation. Existing GTs suffer from noise at both local and global scales. Locally, the input topology exhibits a *sparse and noisy* structure, where structurally inconsistent edges (gray lines); globally, standard attention results in a *dense and noisy* score matrix. Both issues cause noisy inter-cluster propagation. In contrast, the ideal propagation manifests as a *block-diagonal* matrix, where nodes are organized into structurally coherent clusters to suppress structurally heterophilous inter-cluster propagation. (b) Architecture. BDFormer adopts a dual-channel architecture. The Global Channel (bottom) captures long-range dependencies via Bidirectional Node-Anchor Attention, where Spectral Block Regularization is imposed on the latent anchor affinity to encourage a block-diagonal structure. The Local Channel (top) performs Topology Refinement, pruning noisy cross-class edges guided by the learned structure-aware affinity scores for the subsequent GNN. Finally, local and global representations are fused for prediction.

*yields an iterative update that is equivalent to the unified MP. Specifically, the update matches the layer-wise propagation when taking a step size $\eta = 1 - \tau$ and a regularization strength $\lambda = \frac{\tau}{1-\tau}$ (see proof in App. A.1).*

**Motivation for Block-diagonal Priors.** From a graph signal denoising perspective, a block-diagonal propagation structure serves as a structurally favorable prior by suppressing cross-class propagation while promoting intra-class information aggregation. As suggested in Proposition 1, the denoising efficacy of GTs is closely related to the structural quality of $\mathbf{P}$. From this perspective, a desirable propagation operator should encourage structurally coherent information propagation, where signals are primarily exchanged among semantically similar nodes (McPherson et al., 2001), as conceptually illustrated in Fig. 1(a).

However, this structural prior is seldom realized by existing GTs. In unconstrained score-level paradigms (*e.g.*, vanilla dense attention), feature propagation is mediated by dense, all-pair connectivity, which may introduce noisy cross-class propagation between semantically unrelated nodes. In contrast, representation-level paradigms (*e.g.*, GNNs) are strongly constrained by the input graph topology, rendering them vulnerable to topological noise. Such limitations motivate the introduction of an explicit block-diagonal prior, which suppresses noisy cross-class propagation while encouraging coherent intra-class signal propagation.

### 3.2. BDFormer

While the structural prior can be directly imposed via block-diagonal constraints on the $n \times n$ affinity matrix (*e.g.*, BDR (Lu et al., 2018)), such methods are rendered impractical by the $O(n^3)$ cost of matrix factorization. To circumvent this

bottleneck, BDFormer instantiates the structural prior via a set of $k$ latent anchors ($k \ll n$), factorizing the global dependency into efficient bipartite interactions between nodes and latent anchors. By imposing a block-diagonal constraint on these latent assignments, the model achieves linear complexity $O(n)$ while preserving the ability to capture the underlying block-coherent structure, as illustrated in Fig. 1(b).

**Overview.** BDFormer consists of three main modules: (1) Representation Initialization. Latent anchor representations are initialized alongside the projection of node attributes into a shared embedding space. (2) Dual-Channel Propagation. Global node representations are updated via bidirectional node-anchor cross-attention, while the induced affinity patterns are used to refine the input topology for the local GNN branch. (3) Fusion and Prediction. The global and local representations are integrated to produce the final node representations, which are then passed to the prediction layer.

**Node Representations.** Given the input node attributes $\mathbf{X} \in \mathbb{R}^{n \times f}$, the initial node representations $\mathbf{H}^{(0)} \in \mathbb{R}^{n \times d}$ are generated through a parameterized projection, that is,

$$\mathbf{H}^{(0)} = \mathrm{MLP}_{init}(\mathbf{X}), \tag{7}$$

where $\mathrm{MLP}_{init}(\cdot)$ denotes a multi-layer perceptron that maps raw attributes to the $d$-dimensional hidden space.

**Anchor Representations.** To facilitate efficient instantiation of the block-diagonal prior, $k$ latent anchors $\mathbf{B}^{(0)} \in \mathbb{R}^{k \times d}$ are introduced. Here, $k$ is a controlling hyperparameter denoting the number of anchors. Typically, $k > c$ (with $c$ being the number of classes) is set to accommodate potential multi-modal feature distributions within the same class (Caron et al., 2018), thereby capturing finer-grained intra-class structural details. A class-guided anchor initialization

is used to align anchors with semantic clusters, that is,

$$\mathbf{B}^{(0)} = \mathrm{MLP}_{init}(\mathbf{C}) + \mathbf{E}, \tag{8}$$

where $\mathbf{C} \in \mathbb{R}^{k \times f}$ denotes class-wise centroids derived from labeled training nodes. The perturbation matrix $\mathbf{E} \in \mathbb{R}^{k \times d}$ comprises row-wise independent Gaussian noise, introducing diversity among anchors from the same class to prevent representation collapse.

### 3.2.1. DUAL-CHANNEL PROPAGATION

This module synergistically couples global and local channels to update node representations. To be specific, bidirectional node-anchor cross-attention models global dependencies, while a local GNN branch aggregates neighborhood-level information. More importantly, the induced attention patterns are leveraged to refine the input topology, enabling both channels to encourage the block-diagonal prior.

**Bidirectional Node-Anchor Cross-Attention.** To model global dependencies efficiently, the dense interactions are factorized into two steps: (1) Node-to-Anchor (N2A) Aggregation, where anchors collect semantic information from nodes; and (2) Anchor-to-Node (A2N) Broadcasting, where the refined global context is distributed back to nodes.

**Step 1: N2A Aggregation.** This step updates the anchor representations by aggregating relevant features from the nodes. Formally, using anchors $\mathbf{B}$ as Query and nodes $\mathbf{H}$ as Key/Value, the anchor updates $\Delta \mathbf{B}$ are computed as

$$\Delta \mathbf{B} = \mathrm{Attn}(\mathbf{Q}_B, \mathbf{K}_H, \mathbf{V}_H) = \mathbf{S}_{\mathrm{N2A}} \mathbf{V}_H, \tag{9}$$

where $\mathbf{Q}_B = \mathbf{B}\mathbf{W}_{(Q,B)}$, $\mathbf{K}_H = \mathbf{H}\mathbf{W}_{(K,H)}$, and $\mathbf{V}_H = \mathbf{H}\mathbf{W}_{(V,H)}$. And, $\mathbf{S}_{\mathrm{N2A}} \in \mathbb{R}^{k \times n}$ stands for the aggregation weights. Then, the anchors are updated via

$$\mathbf{B}^+ = \mathbf{B}^{(0)} + \Delta \mathbf{B}. \tag{10}$$

**Step 2: A2N Broadcasting.** Utilizing the updated anchors $\mathbf{B}^+$ as the global information source, this step refines the node representations. Nodes $\mathbf{H}$ serve as Query, while updated anchors $\mathbf{B}^+$ act as Key and Value. The node representations are updated as

$$\Delta \mathbf{H} = \mathrm{Attn}(\mathbf{Q}_H, \mathbf{K}_B, \mathbf{V}_B) = \mathbf{S}_{\mathrm{A2N}} \mathbf{V}_B, \tag{11}$$
$$\mathbf{H}_G^+ = \mathbf{H} + \Delta \mathbf{H}, \tag{12}$$

where $\mathbf{S}_{\mathrm{A2N}} \in \mathbb{R}^{n \times k}$ stands for the broadcasting weights distributing global context to individual nodes. Furthermore, a Hard Assignment via Straight-Through Estimator (STE) (Bengio et al., 2013) is adopted to discretize attention scores $\mathbf{S}_{\mathrm{N2A}}$ and $\mathbf{S}_{\mathrm{A2N}}$. This strategy encourages sparse discrete node-anchor assignments, where each node primarily communicates through its most relevant anchor.

**Anchor-Informed Local Aggregation.** To align local feature propagation with the induced block-diagonal structure, the global attention patterns are leveraged to refine the graph topology. Let $\bar{\mathbf{S}}_{\mathrm{N2A}} \in \mathbb{R}^{k \times n}$ and $\bar{\mathbf{S}}_{\mathrm{A2N}} \in \mathbb{R}^{n \times k}$ stand for the layer-averaged node-to-anchor and anchor-to-node attention matrices, respectively. For each observed edge $(i, j) \in \mathcal{E}$, a structure-aware affinity score $w_{ij}$ is computed to measure the structural affinity between nodes via the anchor channel. This can be formulated as

$$w_{ij} = (\bar{\mathbf{S}}_{\mathrm{A2N}})_{i,:} \cdot (\bar{\mathbf{S}}_{\mathrm{N2A}})_{:,j}. \tag{13}$$

This score is computed only for observed edges, ensuring linear complexity $O(|\mathcal{E}|)$. To suppress structurally inconsistent propagation patterns, the topology is refined by retaining a subset of edges $\mathcal{E}' \subset \mathcal{E}$ with relatively high affinity scores, controlled by a sparsity ratio $\rho$. Then, the local representations are updated on this refined topology $\mathbf{A}'$ via

$$\mathbf{H}_L^+ = \mathrm{GNN}(\mathbf{A}', \mathbf{H}). \tag{14}$$

This restricts propagation to block-coherent neighborhoods.

### 3.2.2. FUSION AND PREDICTION

Following the dual-channel propagation, the global node representations $\mathbf{H}_G^+$ and the local representations $\mathbf{H}_L^+$ are integrated to produce the final node representations. A weighted summation controls the trade-off between the two scales:

$$\mathbf{H}^* = \alpha \mathbf{H}_G^+ + (1 - \alpha)\mathbf{H}_L^+, \tag{15}$$

where $\alpha \in [0, 1]$ denotes a tradeoff hyperparameter. The final predictions are generated via a linear projection, namely,

$$\hat{\mathbf{Y}} = \mathrm{softmax}(\mathbf{H}^* \mathbf{W}_{out}), \tag{16}$$

where $\mathbf{W}_{out} \in \mathbb{R}^{d \times c}$ denotes the projection matrix.

### 3.2.3. OBJECTIVE FUNCTION

The training objective combines task-specific supervision with a structural regularization to ensure the learned anchors reflect the desired block-diagonal prior.

**Spectral Block Regularization.** To encourage structurally distinct semantic subspaces, a spectral constraint is imposed on the initialized anchors $\mathbf{B}^{(0)}$. Specifically, an affinity matrix is constructed as $\mathbf{A}_B = \psi\left(\mathbf{B}^{(0)}(\mathbf{B}^{(0)})^\top\right)$, followed by the normalized Laplacian $\mathbf{L}_B = \mathbf{I} - \mathbf{D}^{-1/2}\mathbf{A}_B\mathbf{D}^{-1/2}$. $\psi(\mathbf{M}) = \mathrm{ReLU}\left(\frac{\mathbf{M}+\mathbf{M}^\top}{2}\right) + \epsilon\mathbf{I}$ is a transformation function designed to ensure the affinity matrix is nonnegative and numerically symmetric, which is crucial for stable spectral decomposition. $\epsilon = 10^{-6}$ denotes a small diagonal jitter. $\mathbf{D}$ represents the degree matrix of $\mathbf{A}_B$. Leveraging spectral graph theory (Chung, 1997), minimizing the sum of the $g$

smallest eigenvalues of $\mathbf{L}_B$ promotes the formation of $g$ connected components. This regularization is

$$\mathcal{L}_{\text{spec}} = \sum_{i=1}^{g} \lambda_i\left(\mathbf{L}_B\right), \qquad (17)$$

where $\lambda_i(\cdot)$ stands for the $i$-th smallest eigenvalue.

**Overall Objective.** The loss function augments the cross-entropy classification loss $\mathcal{L}_{\text{task}}$ with the spectral regularization term. This can be formulated as

$$\mathcal{L}_{\text{total}} = \mathcal{L}_{\text{task}} + \beta\mathcal{L}_{\text{spec}}, \qquad (18)$$

where the scalar $\beta$ balances these two terms.

### 3.3. Model Analysis

This subsection analyzes BDFormer from three perspectives: complexity, expressivity, and theoretical properties.

**Complexity.** BDFormer maintains linear time and space complexity with respect to the graph size. Let $n, m, k$ denote the number of nodes, edges, and anchors, respectively, while $h$ and $d$ represent the number of attention heads and the hidden dimension, respectively. **Time Complexity.** The overall complexity is $\mathcal{O}(nkdh + mkh + md)$. This is decomposed into: (1) Global aggregation: $\mathcal{O}(nkdh)$ for bidirectional node-anchor attention; (2) Topology refinement: $\mathcal{O}(mkh)$ for computing affinity scores; (3) Local propagation: $\mathcal{O}(md)$ for GNN aggregation on the refined sparse graph; (4) Spectral regularization: $\mathcal{O}(k^3)$ for anchor eigendecomposition. Note that the $\mathcal{O}(k^3)$ cost is a *small constant overhead*, since typically $k \leq 256 \ll n$. **Space Complexity.** The memory consumption is $\mathcal{O}(nkh + m + nd)$, primarily dominated by the $n \times k$ attention score matrices and the sparse adjacency matrix.

**Expressivity.** The model's representational capacity stems from integrating global receptive fields with structural inductive biases. First, unlike GNNs limited by recursive local aggregation, BDFormer enables direct long-range communication via anchors, effectively circumventing over-squashing. Second, distinct from linear GTs (*e.g.*, NodeFormer (Wu et al., 2022) and SGFormer (Wu et al., 2023)) that approximate dense global interactions, BDFormer imposes a block-diagonal constraint on the global channel. This acts as a structural regularizer, preserving structural distinguishability and mitigating structurally inconsistent cross-class interactions that often compromises dense global attention.

**Effectiveness.** The behavior of BDFormer can be theoretically interpreted through the synergy of spectral block regularization and the STE-based discretization. The former induces a structured latent space, while the latter encourages sparse block-wise interactions.

**Theorem 1** (Anchor-Level Subspace Consistency). *Let $\mathbf{A}_B \in \mathbb{R}^{k \times k}$ be the affinity matrix of the anchors and $\mathbf{L}_B$*

*be its normalized Laplacian. Minimizing the spectral loss $\mathcal{L}_{\text{spec}} = \sum_{i=1}^{g} \lambda_i(\mathbf{L}_B)$ encourages the emergence of approximately $g$ connected components.*

Theorem 1 encourages the global bottleneck to exhibit structured partitioning. To further elucidate how this constraint filters heterophilous noise, the information flow under the discretization strategy is analyzed.

**Proposition 2** (Block-wise Isolation of Global Node-Node Interactions). *Let the global interaction from node $j$ to node $i$ be mediated by the anchor graph via the latent path: Node $j \rightarrow Anchors \rightarrow Node\ i$. Under the STE-based discretization strategy, if the anchor graph is partitioned into disjoint sets $\{\mathcal{C}_1, \ldots, \mathcal{C}_g\}$, and node $i$ and node $j$ exchange features exclusively with anchors in disjoint components, then the effective information flow from $j$ to $i$ via the global channel is effectively suppressed. (Proof in Appendix A.3)*

This suppresses structurally inconsistent cross-class propagation, thereby reducing noisy information mixing.

## 4. Experiments

This section evaluates the effectiveness of BDFormer by answering the following questions: **Q1 (Performance):** How does BDFormer perform on node classification tasks across both homophilic and heterophilic graphs compared to baselines? **Q2 (Scalability):** Can BDFormer scale efficiently to large-scale datasets while maintaining superior performance? **Q3 (Long-Range):** Does the proposed block-diagonal constraint enhance the capability to capture long-range dependencies? **Q4 (Module Analysis):** How do spectral block regularization and topology refinement influence the learned propagation structure? Introductions of the datasets, baseline models, implementation details, and hyperparameter settings are provided in App. B.

### 4.1. Experimental Results

**Performance on Node Classification Tasks (R1).** Table 1 presents the comparison results on eight benchmark datasets encompassing homophilic and heterophilic graphs. In terms of overall performance, BDFormer achieves the most robust results, ranking first in 6 out of 8 datasets with an average rank of $1.25$. On homophilic graphs (*e.g.*, Cora), BDFormer outperforms the majority of GNN and GT baselines, consistently ranking within the top two. This stems from the block-diagonal constraint, which encourages global attention to concentrate on class-consistent dependencies, thereby reducing structurally inconsistent interactions commonly observed in standard GTs. Notably, this superiority is most distinct on heterophilic graphs (*e.g.*, Chameleon, Squirrel). Specifically, BDFormer achieves accuracies of $47.90\%$ and $44.20\%$, significantly outperforming the representation-level paradigm GT SGFormer ($43.77\%$ and $41.09\%$). While

*Table 1.* Accuracy (ACC) or ROC-AUC in percentage (mean$_{\pm \text{std}}$) of the node classification task on homophilic and heterophilic graphs. Best and runner-up models are in **bold** and underlined, respectively.

| Model | Cora | CiteSeer | PubMed | Photo | Computers | Minesweeper | Chameleon | Squirrel | | |
|---|---|---|---|---|---|---|---|---|---|---|
| Metric | ACC ↑ | ACC ↑ | ACC ↑ | ACC ↑ | ACC ↑ | ROC-AUC ↑ | ACC ↑ | ACC ↑ | Avg ↑ | Rank ↓ |
| GCN | $86.53_{\pm1.61}$ | $75.97_{\pm1.93}$ | $88.51_{\pm0.28}$ | $93.07_{\pm0.48}$ | $89.83_{\pm0.64}$ | $93.47_{\pm0.47}$ | $42.87_{\pm2.78}$ | $42.19_{\pm2.10}$ | 76.56 | 12.25 |
| GAT | $86.53_{\pm1.27}$ | $74.31_{\pm1.25}$ | $87.42_{\pm0.43}$ | $93.79_{\pm0.28}$ | $90.45_{\pm0.87}$ | $93.25_{\pm0.42}$ | $41.52_{\pm4.78}$ | $36.59_{\pm1.88}$ | 75.48 | 14.94 |
| GraphSAGE | $87.62_{\pm1.73}$ | $74.79_{\pm1.59}$ | $89.20_{\pm0.53}$ | $94.27_{\pm0.64}$ | $90.14_{\pm0.73}$ | $93.64_{\pm0.39}$ | $42.06_{\pm3.01}$ | $37.21_{\pm1.36}$ | 76.12 | 12.00 |
| GPR-GNN | $88.21_{\pm1.29}$ | $77.02_{\pm1.81}$ | $88.49_{\pm0.44}$ | $94.80_{\pm0.37}$ | $90.70_{\pm0.53}$ | $89.05_{\pm0.43}$ | $41.26_{\pm4.22}$ | $39.80_{\pm1.71}$ | 76.17 | 11.88 |
| GloGNN | $88.17_{\pm1.19}$ | $77.34_{\pm1.08}$ | $88.97_{\pm0.35}$ | $95.11_{\pm0.30}$ | $91.70_{\pm0.45}$ | $93.14_{\pm0.15}$ | $46.13_{\pm2.02}$ | $42.32_{\pm1.92}$ | 77.86 | 6.38 |
| GT | $87.11_{\pm1.07}$ | $77.14_{\pm1.23}$ | $88.76_{\pm0.76}$ | $95.03_{\pm0.43}$ | $90.98_{\pm0.72}$ | $93.47_{\pm0.55}$ | $42.19_{\pm1.31}$ | $41.20_{\pm1.28}$ | 76.99 | 10.19 |
| GraphGPS | $87.02_{\pm1.45}$ | $76.93_{\pm1.21}$ | $89.04_{\pm0.36}$ | $95.16_{\pm0.33}$ | $91.23_{\pm0.72}$ | $95.12_{\pm0.19}$ | $43.73_{\pm1.37}$ | $42.73_{\pm1.30}$ | 77.62 | 8.06 |
| NodeFormer | $87.20_{\pm1.13}$ | $76.80_{\pm1.02}$ | $89.11_{\pm0.50}$ | $95.61_{\pm0.35}$ | $91.24_{\pm0.32}$ | $93.94_{\pm0.19}$ | $42.41_{\pm1.17}$ | $41.20_{\pm1.17}$ | 77.19 | 8.31 |
| NAGphormer | $87.68_{\pm1.80}$ | $76.21_{\pm2.72}$ | $89.35_{\pm0.20}$ | $95.12_{\pm0.36}$ | $91.22_{\pm0.65}$ | $92.10_{\pm0.43}$ | $43.99_{\pm1.90}$ | $41.27_{\pm1.03}$ | 77.12 | 8.75 |
| Exphormer | $87.03_{\pm1.70}$ | $76.18_{\pm1.50}$ | $88.55_{\pm0.47}$ | $95.19_{\pm0.49}$ | $91.29_{\pm0.80}$ | $95.32_{\pm0.93}$ | $41.67_{\pm1.22}$ | $39.82_{\pm1.62}$ | 76.88 | 10.63 |
| GOAT | $87.54_{\pm1.24}$ | $76.44_{\pm1.56}$ | $88.39_{\pm0.49}$ | $95.47_{\pm1.49}$ | $91.63_{\pm0.77}$ | $95.45_{\pm0.33}$ | $42.76_{\pm1.66}$ | $40.81_{\pm1.10}$ | 77.31 | 9.00 |
| SGFormer | $87.86_{\pm1.12}$ | $75.85_{\pm1.04}$ | $88.75_{\pm0.41}$ | $95.10_{\pm0.32}$ | $91.52_{\pm0.68}$ | $91.59_{\pm0.28}$ | $43.77_{\pm3.02}$ | $41.09_{\pm0.73}$ | 76.94 | 10.00 |
| Polynormer | $87.83_{\pm1.94}$ | $76.93_{\pm2.16}$ | $89.48_{\pm0.43}$ | $95.44_{\pm0.71}$ | $91.85_{\pm0.57}$ | $96.98_{\pm0.46}$ | $44.30_{\pm2.04}$ | $41.06_{\pm1.38}$ | 77.98 | 5.69 |
| UGCFormer | $88.01_{\pm0.83}$ | $77.14_{\pm1.52}$ | $\mathbf{89.98}_{\pm0.37}$ | $95.36_{\pm0.57}$ | $\underline{92.22}_{\pm0.39}$ | $96.06_{\pm0.26}$ | $43.72_{\pm1.17}$ | $41.73_{\pm1.89}$ | 78.03 | 4.56 |
| M³Dphormer | $\underline{88.50}_{\pm1.23}$ | $\underline{77.41}_{\pm2.06}$ | $89.91_{\pm0.61}$ | $\underline{96.10}_{\pm0.42}$ | $92.15_{\pm0.36}$ | $\mathbf{97.83}_{\pm0.64}$ | $\underline{47.21}_{\pm2.18}$ | $\underline{43.81}_{\pm1.76}$ | $\underline{79.12}$ | $\underline{2.13}$ |
| BDFormer | $\mathbf{89.01}_{\pm1.13}$ | $\mathbf{77.53}_{\pm1.26}$ | $\underline{89.93}_{\pm0.32}$ | $\mathbf{96.27}_{\pm0.42}$ | $\mathbf{92.41}_{\pm0.27}$ | $\underline{97.35}_{\pm0.17}$ | $\mathbf{47.90}_{\pm1.31}$ | $\mathbf{44.20}_{\pm1.60}$ | **79.33** | **1.25** |

SGFormer fuses global information without structural guidance, BDFormer employs spectral regularization to enforce a block-diagonal affinity. This structurally suppresses noisy cross-class propagation, encouraging the learned representations to better align with the underlying semantic structure.

*Table 2.* ACC or ROC-AUC in percentage (mean$_{\pm \text{std}}$) of property prediction across large-scale graphs. − means unavailable results.

| Model | ogbn-proteins | ogbn-arxiv | ogbn-products |
|---|---|---|---|
| Metric | ROC-AUC ↑ | ACC ↑ | ACC ↑ |
| GCN | $72.51_{\pm0.35}$ | $71.74_{\pm0.29}$ | $75.64_{\pm0.21}$ |
| GAT | $72.02_{\pm0.44}$ | $71.95_{\pm0.36}$ | $79.45_{\pm0.59}$ |
| GraphSAGE | $77.68_{\pm0.20}$ | $71.49_{\pm0.27}$ | $78.50_{\pm0.14}$ |
| GraphGPS | $76.83_{\pm0.26}$ | $70.97_{\pm0.41}$ | $75.39_{\pm0.17}$ |
| NodeFormer | $77.45_{\pm1.15}$ | $67.19_{\pm0.83}$ | $72.93_{\pm0.13}$ |
| NAGphormer | $73.61_{\pm0.33}$ | $70.13_{\pm0.55}$ | $73.55_{\pm0.21}$ |
| Exphormer | $74.58_{\pm0.26}$ | $72.41_{\pm0.40}$ | $76.96_{\pm0.05}$ |
| GOAT | $74.84_{\pm1.16}$ | $72.44_{\pm0.28}$ | - |
| SGFormer | $\underline{79.53}_{\pm0.38}$ | $72.63_{\pm0.13}$ | $74.16_{\pm0.31}$ |
| Polynormer | $78.97_{\pm0.47}$ | $73.46_{\pm0.16}$ | $\underline{83.82}_{\pm0.11}$ |
| UGCFormer | $78.97_{\pm0.47}$ | $\mathbf{74.02}_{\pm0.17}$ | - |
| M³Dphormer | $79.45_{\pm0.75}$ | $73.54_{\pm0.30}$ | - |
| BDFormer | $\mathbf{80.31}_{\pm0.32}$ | $\underline{73.66}_{\pm0.26}$ | $\mathbf{83.98}_{\pm0.14}$ |

**Performance on Large-scale Graphs (R2).** Table 2 reports the results on three large-scale datasets, verifying the model's scalability and effectiveness. Compared to linear GT baselines (*e.g.*, NodeFormer, SGFormer), BDFormer consistently achieves superior performance. Notably, on the dense ogbn-proteins graph, it achieves a state-of-the-art ROC-AUC of $80.31\%$, surpassing the runner-up SGFormer ($79.53\%$). This indicates that the proposed block-coherent propagation effectively captures more structured global dependencies than standard linear attention approximations.

Besides, on ogbn-arxiv, BDFormer remains highly competitive, ranking second with an accuracy of $73.66\%$. These results demonstrate that the proposed BDFormer can successfully scale to large-scale graphs.

*Table 3.* Accuracy (ACC) in percentage (mean$_{\pm \text{std}}$) of node classification on two long-range benchmark (Liang et al., 2025).

| Model | Paris | Shanghai |
|---|---|---|
| Metric | ACC ↑ | ACC ↑ |
| GCN | $53.20_{\pm0.30}$ | $62.10_{\pm0.20}$ |
| GAT | $51.10_{\pm0.30}$ | $68.00_{\pm0.50}$ |
| GraphSAGE | $54.60_{\pm0.20}$ | $68.30_{\pm0.50}$ |
| GT | $54.10_{\pm0.60}$ | $63.00_{\pm0.50}$ |
| GraphGPS | $52.10_{\pm0.60}$ | $63.00_{\pm0.50}$ |
| Exphormer | $55.10_{\pm0.80}$ | $70.20_{\pm0.40}$ |
| SGFormer | $52.00_{\pm0.80}$ | $64.10_{\pm0.30}$ |
| Polynormer | $56.38_{\pm0.57}$ | $\underline{71.42}_{\pm0.46}$ |
| M³Dphormer | $55.29_{\pm0.62}$ | $70.83_{\pm0.39}$ |
| BDFormer | $\mathbf{59.01}_{\pm0.77}$ | $\mathbf{71.69}_{\pm0.51}$ |

**Long-range Modeling Capabilities (R3).** To evaluate the model's ability to capture long-range dependencies, the experiment is conducted on two challenging benchmarks. As shown in Table 3, BDFormer consistently outperforms all baselines, achieving the highest accuracy on both datasets. A critical observation arises from the Paris dataset, a task strictly requiring the identification of distant node relationships. Classic GNNs (*e.g.*, GAT, GCN) perform poorly due to the over-squashing phenomenon inherent in local feature propagation. More surprisingly, global GTs like SGFormer also fail, achieving results comparable to or even worse than GNNs. This indicates that unconstrained global attention tends to distribute attention weights less selectively across nodes, diluting the specific long-range signal needed for the task. In contrast, BDFormer achieves a significant break-

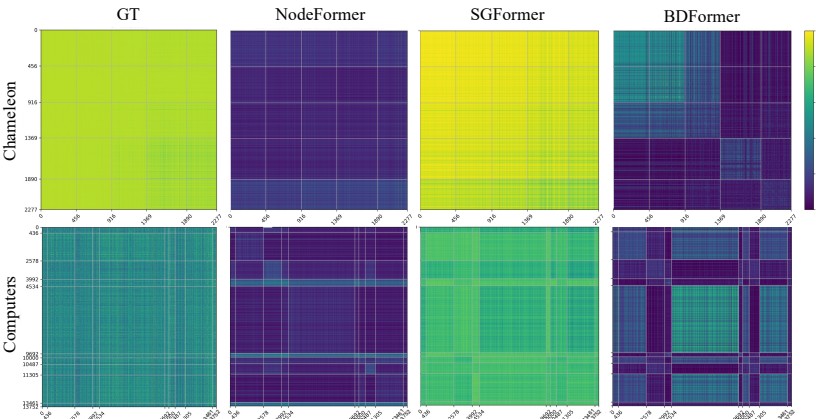

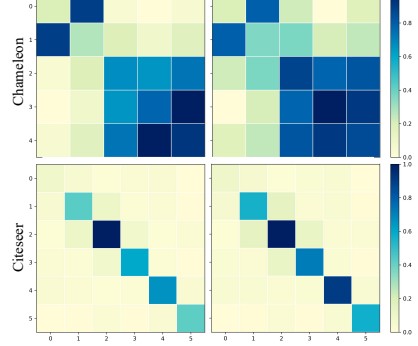

Figure 2. Visual comparison of learned node-to-node attention matrices on heterophilic Chameleon and homophilic Computers. Nodes are ordered according to their labels.

Figure 3. Visualization of inter-class vs. intra-class edge ratios on heterophilic Chameleon and homophilic Citeseer.

through. This suggests that the proposed constraint acts as an effective structural filter, enabling precise long-range propagation where unconstrained global attention fails.

### 4.2. Additional Analysis

**Module Analysis (R4).** The impact of the spectral block regularization and topology refinement is investigated by visualizing the learned attention patterns and graph topology evolution (Input vs. Pruned) in Fig. 2 and Fig. 3, respectively. **Global Attention Patterns.** Fig. 2 contrasts the learned attention maps. Standard global GTs (*e.g.*, SG-Former) exhibit diffuse, uniform distributions, struggling to capture distinct block-diagonal structures. Moreover, Node-Former reveals a fundamental limitation: while it forms diagonal patterns on homophilic Computers, it becomes a liability on heterophilic Chameleon. By relying heavily on topology-aware attention patterns, NodeFormer may unintentionally reinforce noisy cross-class interactions (visible as scattered off-diagonal weights). In contrast, BDFormer exhibits clearer block-diagonal patterns compared to baselines. Even on the heterophilic Chameleon, it demonstrates a stronger tendency to concentrate attention into diagonal blocks, proving that the spectral constraint encourages more structurally coherent propagation blocks. **Topology Refinement.** Fig. 3 verifies the pruning effectiveness by comparing class-wise connectivity distributions. The heatmaps of the input graph, particularly for Chameleon, display strong off-diagonal density, reflecting severe heterophily. Conversely, the heatmaps of the pruned graph demonstrate a marked structural shift. A rigorous block-by-block inspection confirms effective off-diagonal suppression and diagonal enhancement: for Chameleon, off-diagonal blocks (*e.g.*, (0,1), (1,0), (3,4), (4,3)) become visibly lighter, while diagonal blocks (*e.g.*, (1,1), (3,3)) deepen in color. A similar topological enhancement is observed on CiteSeer, with diagonal blocks (*e.g.*, (2,2), (3,3), (4,4), (5,5)) distinctly intensifying. This suggests that the pruning mechanism functions

as an effective structural denoising mechanism, effectively enhancing structural homophily.

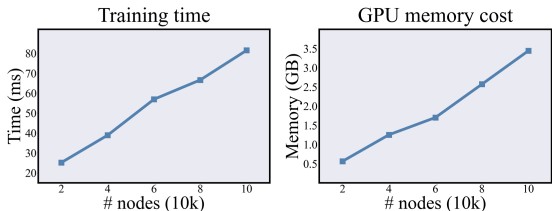

Figure 4. Empirical verification of linear scalability.

**Complexity and Scalability Verification.** To empirically validate the theoretical linear complexity of BDFormer, a controlled scalability test is conducted on the ogbn-arxiv dataset. In this experimental setup, model hyperparameters (*e.g.*, depth and hidden dimension) are kept fixed, while the input subgraph size is varied from 20K to 100K nodes to monitor resource consumption. As illustrated in Fig. 4, both the average training time per epoch and GPU memory usage exhibit a linear growth trend with respect to the number of nodes. The resource consumption increases proportionally without the quadratic explosion observed in standard Transformers. This empirical evidence is consistent with the complexity analysis in Section 3.3, highlighting that BD-Former scales linearly and remains computationally efficient for processing large-scale graphs.

**Hyper-parameter Sensitivity Analysis.** To provide practical guidance for model deployment, the impact of the micro-level anchors ($k$) and meso-level blocks ($g$) is investigated. Conceptually, setting $k > c$ provides an over-complete basis to capture the multi-modal feature distributions within each macro-level semantic class $c$, while $g$ functions as a structural bottleneck that aggregates these micro-anchors via spectral regularization. As illustrated in Fig. 5, BD-Former exhibits remarkable performance stability across a wide range of these parameter choices. While increasing $k$

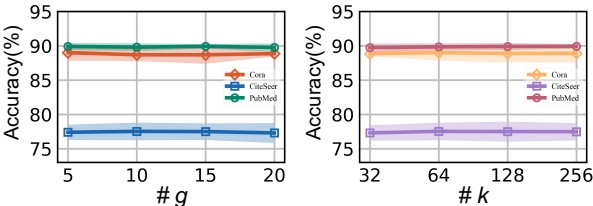

*Figure 5.* Performance variation with varying $k$ and $g$.

or $g$ initially allows for more expressive structural modeling, the accuracy gains saturate once $k \geq 64$ and $g \geq 10$. This confirms that BDFormer robustly captures the dominant propagation structure even with moderate structural sizes, eliminating the need for exhaustive hyperparameter tuning. Evaluations on the regularization strength ($\beta$) are provided in App. C.3, further verifying the model's stability.

## 5. Conclusions and Limitations

This paper presents BDFormer, a novel Graph Transformer designed to capture latent global structures while maintaining strict linear complexity. Unlike vanilla Transformers that suffer from quadratic overhead, the model introduces a discrete anchor-based mechanism to efficiently partition global interactions into a structured format. By integrating a block-diagonal prior, it effectively suppresses structurally noisy cross-class propagation and prunes redundant connections, thereby refining feature propagation in both homophilic and heterophilic graphs. **Limitations.** While BDFormer achieves advanced performances, its structural optimization benefits from class-guided anchor initialization, implying a dependency on label availability. A natural limitation arises in scenarios where such prior label information is strictly unavailable for initialization. In these cases, the anchor initialization can adopt standard unsupervised K-means over the input node attributes; empirical evaluations demonstrate that even with this label-free initialization, driven by the robust structural prior enforced by the proposed spectral constraint, BDFormer does not suffer from representation collapse and maintains highly competitive performance (see App. C.5). **Future Work.** Several directions remain to further advance this architecture. First, extending the fixed anchor set to an adaptive anchor allocation mechanism could optimize structural representations. Moreover, adapting the block-diagonal prior to handle heterogeneous graphs with diverse relation types presents a valuable avenue for cross-domain applications.

## Acknowledgements

This work was supported in part by the National Natural Science Foundation of China (No. 92570118, U22B2036, 62376088, 62272020, 62025604, 92370111, 62272340, 62261136549, 52441501), in part by the Hebei Natural Science Foundation (No. F2024202047), in part by the National Science Fund for Distinguished Young Scholarship (No. 62025602), in part by the Hebei Yanzhao Golden Platform Talent Gathering Programme Core Talent Project (Education Platform) (HJZD202509), in part by the Post-graduate's Innovation Fund Project of Hebei Province (CXZZBS2025036), in part by the Tencent Foundation, and in part by the XPLORER PRIZE.

## Impact Statement

This paper presents work whose goal is to advance the field of Machine Learning. There are many potential societal consequences of our work, none of which we feel must be specifically highlighted here.

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

---

**Algorithm 1:** BDFormer Training Algorithm

---

**Input:** Graph $\mathcal{G}(\mathbf{A}, \mathbf{X})$, Labels $\mathbf{Y}$, Hyperparams $\rho, g, k, \ell_G, \ell_L, \alpha, \beta$
**Output:** Optimized parameters $\Theta^*$
**Initialization:** Randomly initialize network parameters $\Theta$

1 **while** *not converged* **do**
  // Phase 1:  Initialization
2     Generate initial node representations $\mathbf{H}^{(0)}$ via Eq. 7;
3     Generate initial anchor representations $\mathbf{B}^{(0)}$ via Eq. 8;
  // Phase 2:  Global Channel
4     **for** $\ell = 1$ **to** $\ell_G$ **do**
5        Update anchors $\mathbf{B}^{(\ell)}$ via N2A attention (Eqs. 9 and 10);
6        Update nodes $\mathbf{H}_G^{(\ell)}$ via A2N attention (Eqs. 11 and 12);
  // Phase 3:  Topology Refinement
7     Compute structure-aware affinity scores $w_{ij}$ via Eq. 13;
8     Refine topology $\bar{\mathbf{A}}$ by suppressing structurally inconsistent edges (Top-$\rho$);
  // Phase 4:  Local Channel
9     **for** $\ell = 1$ **to** $\ell_L$ **do**
10        Update local node representations $\mathbf{H}_L^{(\ell)}$ on $\bar{\mathbf{A}}$ via Eq. 14;
  // Phase 5:  Fusion and Optimization
11     Obtain final prediction $\hat{\mathbf{Y}}$ via Eqs. 15 and 16;
12     Compute overall loss $\mathcal{L}_{\text{total}} = \mathcal{L}_{\text{task}} + \beta\mathcal{L}_{\text{spec}}$ via Eqs. 17 and 18;
13     Update $\Theta \leftarrow \text{Adam}(\mathcal{L}_{\text{total}}, \Theta)$;
14 **return** *Parameters* $\Theta^*$

---

# A. Theoretical Analysis and Proofs

### A.1. Proof of Proposition 1

**Proof.** The graph signal denoising objective is defined as:

$$\mathcal{L}(\mathbf{Z}) = \frac{1}{2}\|\mathbf{Z} - \mathbf{H}^{(0)}\|_F^2 + \frac{\lambda}{2}\operatorname{tr}\left(\mathbf{Z}^\top(\mathbf{I} - \mathbf{P})\mathbf{Z}\right). \tag{19}$$

Taking the derivative of the objective function $\mathcal{L}(\mathbf{Z})$ with respect to the variable $\mathbf{Z}$, the gradient of the first term is exactly $\nabla_{\mathbf{Z}}\mathcal{L}_1 = \mathbf{Z} - \mathbf{H}^{(0)}$. For the second term, since the affinity matrix $\mathbf{P}$ in GTs is typically asymmetric (*e.g.*, due to softmax distributions), the exact gradient of the quadratic form would symmetrize the operator as $\mathbf{I} - \frac{1}{2}(\mathbf{P} + \mathbf{P}^\top)$.

To preserve the directed propagation behavior induced by $\mathbf{P}$, we adopt the directed operator $\mathbf{I} - \mathbf{P}$ in the update rule. Following the principles of directed graph diffusion, we adopt the directed propagation operator $\mathbf{I} - \mathbf{P}$, thereby preserving the structural directionality. Combining these, the resulting directed diffusion update is formulated as

$$\nabla\mathcal{L}(\mathbf{Z}) := (\mathbf{Z} - \mathbf{H}^{(0)}) + \lambda(\mathbf{I} - \mathbf{P})\mathbf{Z}. \tag{20}$$

During the forward propagation of the model, applying gradient descent to optimize the graph signal state $\mathbf{Z}^{(l-1)}$ at the $l$-th iteration with a step size (learning rate) $\eta$ yields the following update rule:

$$\mathbf{Z}^{(l)} = \mathbf{Z}^{(l-1)} - \eta\nabla\mathcal{L}(\mathbf{Z}^{(l-1)}). \tag{21}$$

Substituting the directed gradient into the above equation and regrouping the terms, we obtain:

$$\begin{aligned}
\mathbf{Z}^{(l)} &= \mathbf{Z}^{(l-1)} - \eta\left[(\mathbf{Z}^{(l-1)} - \mathbf{H}^{(0)}) + \lambda(\mathbf{I} - \mathbf{P})\mathbf{Z}^{(l-1)}\right] \\
&= \eta\mathbf{H}^{(0)} + [1 - \eta(1 + \lambda)]\mathbf{Z}^{(l-1)} + \eta\lambda\mathbf{P}\mathbf{Z}^{(l-1)}.
\end{aligned} \tag{22}$$

To establish a strict equivalence between this energy-based optimization step and the GT feature propagation paradigm governed by the hyperparameter $\tau$, we construct a parameter mapping. By setting the optimization step size $\eta = 1 - \tau$ and configuring the regularization strength to satisfy $\eta\lambda = \tau$ (which implies the implicit structural prior $\lambda = \frac{\tau}{1-\tau}$), we substitute this mapping into the expanded formulation. Substituting the parameterization yields: $1 - \eta(1 + \lambda) = 1 - \eta - \eta\lambda = 1 - (1 - \tau) - \tau = 0$. The update becomes:

$$\mathbf{Z}^{(l)} = (1 - \tau)\mathbf{H}^{(0)} + \tau\mathbf{P}\mathbf{Z}^{(l-1)}. \tag{23}$$

This derived result is algebraically identical to the unified message passing update (Eq. 5). This establishes that the unified message passing update can be interpreted as a directed iterative approximation for minimizing the associated graph signal denoising objective. ∎

### A.2. Proof of Theorem 1

*Proof.* The normalized Laplacian $\mathbf{L}_B$ is symmetric and positive semi-definite, implying that all its eigenvalues are non-negative. Without loss of generality, let the eigenvalues be sorted as $0 \leq \lambda_1 \leq \lambda_2 \leq \cdots \leq \lambda_k$. The objective function is defined as the sum of the $g$ smallest eigenvalues, namely,

$$\mathcal{L}_{\text{spec}} = \sum_{i=1}^{g} \lambda_i(\mathbf{L}_B). \tag{24}$$

Since $\lambda_i \geq 0$, the condition $\mathcal{L}_{\text{spec}} = 0$ strictly implies that $\lambda_1 = \lambda_2 = \cdots = \lambda_g = 0$.

According to classic Spectral Graph Theory (Chung, 1997), the multiplicity of the eigenvalue $0$ of the normalized Laplacian is exactly equal to the number of connected components in the graph. Therefore, the existence of $g$ zero eigenvalues implies that the anchor graph consists of $p$ disjoint connected components, where $p \geq g$. Let these components be denoted as $\{\mathcal{C}_1, \mathcal{C}_2, \ldots, \mathcal{C}_p\}$.

In matrix form, this implies there exists a permutation matrix $\mathbf{\Pi}$ such that the permuted affinity matrix $\mathbf{\Pi}\mathbf{A}_B\mathbf{\Pi}^\top$ is block-diagonal, that is,

$$\mathbf{\Pi}\mathbf{A}_B\mathbf{\Pi}^\top = \text{diag}(\mathbf{B}_1, \mathbf{B}_2, \ldots, \mathbf{B}_p), \tag{25}$$

where each block $\mathbf{B}_m$ represents the internal connectivity of the component $\mathcal{C}_m$, and all off-diagonal blocks are zero matrices. This completes the proof. □

### A.3. Proof of Proposition 2

*Proof.* In the global channel of BDFormer, the information exchange is mediated by the anchors. Mathematically, the effective weight $w_{i,j}$ representing the information flow from a source node $j$ to a target node $i$ corresponds to the path integration via the anchors, namely, $w_{i,j} = (\mathbf{S}_{\text{A2N}})_{i,:} \cdot (\mathbf{S}_{\text{N2A}})_{:,j} = \sum_{r=1}^{k}(\mathbf{S}_{\text{A2N}})_{i,r} \cdot (\mathbf{S}_{\text{N2A}})_{r,j}$. Here, $\mathbf{S}_{\text{N2A}} \in \mathbb{R}^{k \times n}$ (rows indexed by anchors, columns by nodes) and $\mathbf{S}_{\text{A2N}} \in \mathbb{R}^{n \times k}$ (rows indexed by nodes, columns by anchors) denote the discretized attention matrices derived from the STE mechanism.

Assume the anchor space is partitioned into disjoint connected components $\{\mathcal{C}_1, \ldots, \mathcal{C}_g\}$, a structure induced by the Spectral Block Regularization. Let node $i$ be associated with component $\mathcal{C}_a$ and node $j$ be associated with component $\mathcal{C}_b$, with $a \neq b$.

The premise that nodes exchange features exclusively with anchors in their respective components implies:

- For the target node $i$: $(\mathbf{S}_{\text{A2N}})_{i,r} \neq 0$ only if anchor $r \in \mathcal{C}_a$.

- For the source node $j$: $(\mathbf{S}_{\text{N2A}})_{r,j} \neq 0$ only if anchor $r \in \mathcal{C}_b$.

The term $T_r = (\mathbf{S}_{\text{A2N}})_{i,r} \cdot (\mathbf{S}_{\text{N2A}})_{r,j}$ is analyzed for any arbitrary anchor $r \in \{1, \ldots, k\}$: (1) If $r \in \mathcal{C}_a$, then $r \notin \mathcal{C}_b$ (since $\mathcal{C}_a \cap \mathcal{C}_b = \emptyset$). Consequently, the aggregation weight from node $j$ is zero, *i.e.*, $(\mathbf{S}_{\text{N2A}})_{r,j} = 0$, leading to $T_r = (\mathbf{S}_{\text{A2N}})_{i,r} \cdot 0 = 0$. (2) If $r \in \mathcal{C}_b$, then $r \notin \mathcal{C}_a$. Consequently, the broadcasting weight to node $i$ is zero, *i.e.*, $(\mathbf{S}_{\text{A2N}})_{i,r} = 0$, leading to $T_r = 0 \cdot (\mathbf{S}_{\text{N2A}})_{r,j} = 0$. (3) If $r$ belongs to neither $\mathcal{C}_a$ nor $\mathcal{C}_b$, both terms are zero, leading to $T_r = 0$.

Since the term $T_r$ is strictly zero for all anchors $r$, the total effective information flow $w_{i,j} = \sum_{r=1}^{k} T_r = 0$. This completes the proof. □

# B. Experimental Details

**Datasets.** Experiments are conducted on 13 datasets (see Tab. 4) categorized into four distinct groups based on their structural properties and scales:

- **Standard Homophilic and Structured Benchmarks:** This group comprises five widely used homophilic graphs, including three citation networks (Cora, CiteSeer, PubMed) (Sen et al., 2008) and two co-purchase networks (Amazon-Photo, Amazon-Computers) (Shchur et al., 2018). Additionally, the synthetic Minesweeper dataset is incorporated.

- **Heterophilic Benchmarks:** To assess performance under low-homophily conditions, the filtered versions of Chameleon and Squirrel (denoted with (*filt.*) in Tab. 4) (Platonov et al., 2023) are employed. These versions are adopted to eliminate known leakage issues.

- **Large-scale OGB Benchmarks:** Scalability is verified using ogbn-arxiv, ogbn-proteins, and ogb-products from the Open Graph Benchmark (OGB) (Hu et al., 2020). These datasets represent significant challenges due to their substantial node counts and edge densities, providing a robust stress test for computational efficiency.

- **Long-range Benchmarks:** The Paris and Shanghai datasets (Liang et al., 2025) are utilized to evaluate the model's capacity for capturing long-range dependencies.

*Table 4.* Statistics of target graph datasets. $\#h$ denotes the edge homophily shown in (Pei et al., 2020).

| Dataset | Nodes | Edges | Features | Classes | $\#h$ |
|---|---|---|---|---|---|
| Cora | 2,708 | 5,278 | 1,433 | 7 | 0.81 |
| CiteSeer | 3,327 | 4,552 | 3,703 | 6 | 0.74 |
| PubMed | 19,717 | 44,324 | 500 | 3 | 0.80 |
| Photo | 7,650 | 238,162 | 745 | 8 | 0.84 |
| Computers | 13,752 | 245861 | 767 | 10 | 0.79 |
| Minesweeper | 10,000 | 39,402 | 7 | 2 | 0.68 |
| Chameleon (*filt.*) | 2,277 | 31,421 | 2,325 | 5 | 0.23 |
| Squirrel (*filt.*) | 5,201 | 198,493 | 2,089 | 5 | 0.22 |
| ogbn-Arxiv | 169,343 | 1,166,243 | 300 | 40 | 0.65 |
| ogbn-proteins | 132,534 | 39,561,252 | 8 | 2 | 0.38 |
| ogbn-products | 2,449,029 | 61,859,140 | 100 | 47 | 0.81 |
| Paris | 114,127 | 182511 | 12 | 10 | 0.70 |
| Shanghai | 183,917 | 262,092 | 12 | 10 | 0.75 |

**Dataset Splitting.** This paper adheres to the dataset splitting strategies strictly consistent with the baselines (Wu et al., 2022; Xing et al., 2025) and established benchmarks (Hu et al., 2020). For the Amazon-Photo and Amazon-Computers datasets, nodes are randomly divided into training, validation, and testing sets with a $60\% : 20\% : 20\%$ ratio. For the large-scale OGB datasets, *i.e.*, ogbn-arxiv and ogbn-proteins, the official splits (Hu et al., 2020) are utilized. The remaining benchmarks, including the citation networks (Cora, CiteSeer, and PubMed), the Wikipedia networks (Chameleon and Squirrel), and the synthetic Minesweeper dataset, are randomly split into training, validation, and testing sets using a $50\% : 25\% : 25\%$ ratio. Finally, following the experimental protocol in (Liang et al., 2025), transductive node classification on the Shanghai and Paris datasets is performed using a $10\% : 10\% : 80\%$ train/validation/test split.

## B.1. Introduction of Baselines

To rigorously evaluate the effectiveness of BDFormer, the comparative analysis involves a total of 15 baseline models, comprising 5 Graph Neural Networks (GNNs) and 10 state-of-the-art Graph Transformers (GTs). These models are introduced as follows.

### B.1.1. GRAPH NEURAL NETWORKS

- **GCN** (Kipf & Welling, 2017), **GAT** (Velickovic et al., 2017), and **GraphSAGE** (Hamilton et al., 2017): These represent classic message-passing architectures. **GCN** utilizes graph convolutions for neighborhood aggregation, **GAT** introduces an attention mechanism to assign weights to edges, and **GraphSAGE** achieves scalability on large-scale graphs through neighbor sampling techniques.

- **GPR-GNN** (Chien et al., 2021): A universal GNN that employs Generalized PageRank with learnable coefficients to adaptively handle diverse homophily levels.

- **GloGNN** (Li et al., 2022): A global GNN that captures multi-hop dependencies by aggregating information from all nodes, where the aggregation coefficients are derived from a closed-form solution to an optimization objective.

### B.1.2. GRAPH TRANSFORMERS

- **GT** (Dwivedi & Bresson, 2020): A classic architecture that generalizes Transformers to graphs by treating nodes as a sequence, relying on Laplacian positional encodings to preserve global topology information.

- **GraphGPS** (Rampásek et al., 2022): A modular hybrid GT that decouples local structural message-passing from linear global attention, integrated with diverse positional and structural encodings.

- **NodeFormer** (Wu et al., 2022): A scalable GT that enables all-pair message passing via a kernelized Gumbel-Softmax operator, allowing for the differentiable learning of layer-specific latent graph structures with linear complexity.

- **NAGphormer** (Chen et al., 2023): A tokenized GT that treats each node as a sequence of tokens via a Hop2Token module, utilizing a Hop-wise Attention mechanism to aggregate multi-hop neighborhood features.

- **Exphormer** (Shirzad et al., 2023): A GT that achieves efficient global connectivity with linear complexity by constructing sparse attention patterns based on expander graphs, virtual nodes, and local graph structures.

- **GOAT** (Kong et al., 2023): A GT that achieves linear complexity in all-node attention by utilizing k-means clustering as a Fast Global Self-attention scheme.

- **SGFormer** (Wu et al., 2023): A simplified GT featuring a one-layer global attention module to capture all-pair interactions, specifically designed to scale to large graphs with linear complexity.

- **Polynormer** (Deng et al., 2024): A polynomial-expressive GT with linear complexity that learns high-degree feature interactions by integrating local and global equivariant attention modules.

- **UGCFormer** (Zhuo et al., 2025b): A universal GT that employs a cross-aggregation mechanism to adaptively integrate graph topology and node attributes.

- **M$^3$Dphormer** (Xing et al., 2025): A modular GT that integrates diverse node interactions by leveraging a hierarchical mask framework and a Mixture-of-Masked-Experts mechanism.

All baseline models are implemented utilizing their officially released source code to ensure consistent hyperparameter tuning and experimental conditions. For the classic GNNs—specifically GCN, GAT, and GraphSAGE—we employ the standard PyTorch Geometric (PyG) library (Fey & Lenssen, 2019). For more specialized GNNs such as GPR-GNN and GloGNN, as well as the entire suite of GT baselines, we strictly adhere to the original repositories and experimental protocols provided by the respective authors. The specific source links for these implementations are summarized as follows

- GCN: `https://github.com/tkipf/gcn`

- GAT: `https://github.com/PetarV-/GAT`

- GraphSAGE: `https://github.com/williamleif/GraphSAGE`

- GPR-GNN: `https://github.com/jianhao2016/GPRGNN`

- GloGNN: `https://github.com/THUDM/GloGNN`

For the GT baselines, including GraphGPS, NodeFormer, NAGphormer, Exphormer, GOAT, SGFormer, Polynormer, and Gradformer, we utilize their source code. The sources are detailed as

- GT: `https://github.com/graphdeeplearning/graphtransformer`

- GraphGPS: https://github.com/rampasek/GraphGPS

- NodeFormer: https://github.com/qitianwu/NodeFormer

- NAGphormer: https://github.com/JHL-HUST/NAGphormer

- Exphormer: https://github.com/hamed1375/Exphormer

- GOAT: https://github.com/devnkong/GOAT

- SGFormer: https://github.com/qitianwu/SGFormer

- Polynormer: https://github.com/cornell-zhang/Polynormer

- UGCFormer: https://openreview.net/attachment?id=7FhWZFoVem&name=supplementary_material

- M$^3$Dphormer: https://github.com/null-xyj/M3Dphormer

## B.2. Experimental Setups

### B.2.1. HARDWARE AND PROTOCOL

The experiments are conducted on two Linux workstations, equipped with an NVIDIA GeForce RTX 4090 (24 GB) and an NVIDIA A800 (80 GB) GPU, respectively. All models are implemented using the PyTorch framework and evaluated under a semi-supervised learning paradigm. To ensure statistical reliability, the reported results represent the average performance over five independent trials with varied random seeds.

### B.2.2. IMPLEMENTATIONS AND HYPER-PARAMETERS

All models undergo an extensive grid search to identify optimal configurations. The search space is organized from general optimization settings to model-specific components. Regarding common optimization, the Adam optimizer is employed with a fixed learning rate $lr$ of $0.001$, while the weight decay $wd$ is searched within $\{0, 5 \times 10^{-5}, 5 \times 10^{-4}\}$. For architectural settings, the hidden dimension is selected from $\{64, 128, 256\}$, the dropout rate is set to $0.5$, the numbers of GNN and attention layers are tuned within $\{1, 2, 3, 4\}$, and the number of heads is fixed at $4$. Parameters unique to specific baselines that fall outside this common space follow the values recommended in their original papers. For the specific designs of our BDFormer, the number of anchors ($k$) is searched in $\{32, 64, 128, 256\}$ and the number of blocks ($g$) in $\{5, 10, 15, 20\}$. The spectral coefficient $\beta$ is tuned within $\{0.01, 0.1, 1\}$ and the edge preservation proportion ($\rho$) for topology pruning is fixed at $0.8$. The trade-off parameter $\alpha$, balancing local and global scales, is searched across $\{0.1, 0.2, \dots, 0.9\}$. Additionally, BDFormer is a modular framework compatible with diverse GNN backbones. For the node classification and large-scale tasks, SGC-style linear propagation (Wu et al., 2019) is adopted for efficiency. For the long-range dependency tasks, a GCN-based configuration (Luo et al., 2024) is employed with its depth extended to 16 layers, following established protocols (Liang et al., 2025).

## C. Additional Experiments

### C.1. Ablation Study

To evaluate the contributions of discrete assignment and spectral regularization, BDFormer with two variants, namely, a soft-assignment version and a version without spectral loss is compared (Tab. 5). **Impact of Discrete Assignment via STE**. Compared to the continuous (soft) variant, the default hard assignment yields consistent improvements, particularly on heterophilic datasets such as Chameleon ($+1.14\%$) and Squirrel ($+0.79\%$). Although discrete operations are inherently non-differentiable, the STE effectively mitigates this by enabling stable gradient flow during backpropagation. This method allows the model to leverage the benefits of structural sparsification—effectively filtering out heterophilic noise and emphasizing task-relevant interactions—which a soft, dense weight assignment fails to achieve. **Effectiveness of Spectral Regularization**. Removing the spectral loss (that is, *w/o* $\mathcal{L}_{\text{spec}}$) leads to performance degradation across all datasets. The consistent degradation across datasets highlights the role of spectral regularization in stabilizing the learned structures. By enforcing theoretical consistency with the intrinsic global graph topology, the spectral loss prevents over-parameterization and ensures that the refined structure remains robust and interpretable.

*Table 5.* Accuracy (ACC) or ROC-AUC in percentage (mean$_{\pm\text{std}}$) of the node classification task on homophilic and heterophilic graphs. Best and runner-up models are in **bold** and underlined, respectively.

| Model
Metric | Cora
ACC ↑ | CiteSeer
ACC ↑ | PubMed
ACC ↑ | Photo
ACC ↑ | Computers
ACC ↑ | Minesweeper
ROC-AUC ↑ | Chameleon
ACC ↑ | Squirrel
ACC ↑ |
|---|---|---|---|---|---|---|---|---|
| BDFormer | **89.01**$_{\pm1.13}$ | 77.53$_{\pm1.26}$ | **89.93**$_{\pm0.32}$ | 96.27$_{\pm0.42}$ | **92.41**$_{\pm0.27}$ | **97.35**$_{\pm0.17}$ | **47.90**$_{\pm1.31}$ | **44.20**$_{\pm1.60}$ |
| BDFormer (Soft) | 89.00$_{\pm1.17}$ | **77.55**$_{\pm1.33}$ | 89.85$_{\pm0.37}$ | **96.30**$_{\pm0.39}$ | 92.30$_{\pm0.27}$ | 97.33$_{\pm0.20}$ | 46.76$_{\pm1.50}$ | 43.41$_{\pm1.29}$ |
| BDFormer *w/o* $\mathcal{L}_{\text{spec}}$ | 88.30$_{\pm0.41}$ | 77.14$_{\pm1.19}$ | 89.91$_{\pm0.40}$ | 96.15$_{\pm0.37}$ | 92.01$_{\pm0.31}$ | 97.01$_{\pm0.22}$ | 46.31$_{\pm1.43}$ | 42.75$_{\pm1.36}$ |

## C.2. Efficiency and Scalability Analysis

Figure 6 shows the trade-offs between accuracy, training time, and GPU memory usage. BDFormer consistently occupies the top-left frontier, demonstrating superior efficiency. While heavy models like GraphGPS and Exphormer suffer from high latency, BDFormer achieves competitive or superior accuracy with significantly lower time complexity. Compared to ultra-lightweight models like SGFormer, BDFormer offers substantial accuracy gains with only marginal temporal overhead. Besides, its compact bubble size reflects optimized memory efficiency. This stems from the anchor-based mechanism and linear backbone, which bypass the quadratic bottleneck of attention while effectively capturing long-range dependencies.

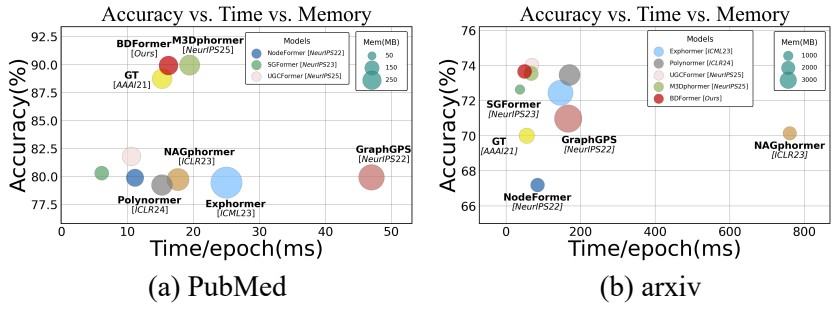

(a) PubMed  (b) arxiv

*Figure 6.* Accuracy vs. time vs. memory comparison between GTs.

## C.3. Additional Hyper-parameter Sensitivity Analysis

To provide practical guidance for hyper-parameter selection, how model performance varies with architectural depth and the spectral coefficient ($\beta$) is analyzed. The selected parameters corresponding to the results of each dataset are shown in Tab. 6.

**Layer Depths.** As illustrated in Fig. 7, BDFormer exhibits strong robustness to depth variations across diverse datasets. For GNN components, performance typically peaks at 3 or 4 layers, suggesting that moderate depth effectively captures local structural information while avoiding over-smoothing. Similarly, for attention layers, a depth of 2 or 3 consistently yields near-optimal performance across all benchmarks. This indicates that a relatively shallow global interaction mechanism is sufficient to capture long-range dependencies while maintaining high computational efficiency.

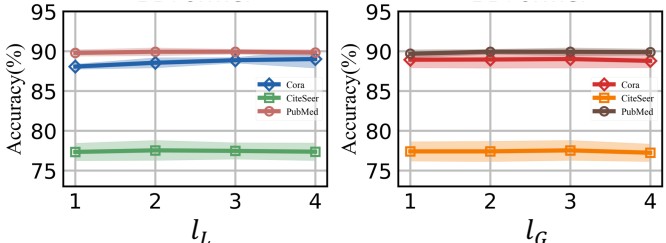
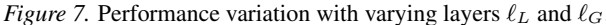
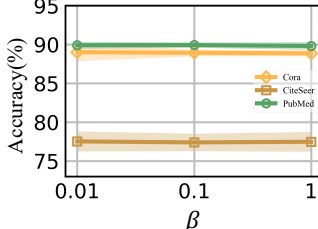

*Figure 7.* Performance variation with varying layers $\ell_L$ and $\ell_G$.

*Figure 8.* Performance variation with varying $\beta$.

**Effect of Spectral Coefficient ($\beta$).** The spectral coefficient $\beta$ controls the strength of structural regularization. As shown in Fig. 8, the model achieves optimal performance when $\beta$ is within the range of $\{0.01, 0.1\}$. While an insufficiently small $\beta$ may weaken the structural constraints, an excessively large value can potentially overshadow the primary task loss, leading to slight performance drops. The observed stability within this range indicates that the spectral loss provides a reliable regularization effect that is robust to precise parameter tuning.

*Table 6.* Selected hyperparameters of BDFormer per dataset.

| Dataset | $\ell_L$ | $\ell_G$ | $k$ | $g$ | $\beta$ | $wd$ | hidden | $\alpha$ |
|---|---|---|---|---|---|---|---|---|
| Cora | 4 | 3 | 64 | 5 | 0.01 | 0 | 128 | 0.4 |
| CiteSeer | 2 | 3 | 64 | 10 | 0.01 | $5 \times 10^{-5}$ | 256 | 0.3 |
| PubMed | 3 | 2 | 256 | 15 | 0.1 | $5 \times 10^{-5}$ | 256 | 0.4 |
| Photo | 2 | 4 | 256 | 10 | 0.01 | 0 | 256 | 0.2 |
| Computers | 4 | 3 | 128 | 15 | 0.1 | $5 \times 10^{-4}$ | 256 | 0.4 |
| Minesweeper | 2 | 3 | 128 | 10 | 0.01 | $5 \times 10^{-5}$ | 128 | 0.3 |
| Chameleon | 4 | 4 | 32 | 10 | 0.1 | $5 \times 10^{-5}$ | 128 | 0.8 |
| Squirrel | 2 | 3 | 64 | 5 | 0.1 | $5 \times 10^{-4}$ | 256 | 0.6 |
| ogbn-Arxiv | 2 | 2 | 256 | 5 | 0.01 | $5 \times 10^{-4}$ | 256 | 0.3 |
| ogbn-proteins | 2 | 3 | 128 | 10 | 0.01 | $5 \times 10^{-4}$ | 256 | 0.2 |
| Paris | 16 | 4 | 128 | 10 | 0.01 | $5 \times 10^{-4}$ | 256 | 0.1 |
| Shanghai | 16 | 4 | 256 | 15 | 0.01 | $5 \times 10^{-5}$ | 256 | 0.2 |

## C.4. Comparison with Polynormer under Matched Configuration Spaces

This experiment aims to provide an additional comparison between BDFormer and Polynormer under the exact resource constraints originally used in the Polynormer paper. The best results reported by Polynormer were achieved using a massive architectural space. Accordingly, we aligned our model capacity by adopting their configuration ranges: a hidden dimension of 512, 8 attention heads, dropout rates selected from $\{0.2, 0.3, 0.5, 0.7\}$, and network depths spanning up to 7 layers.

As shown in Tab. 7, BDFormer consistently outperforms Polynormer across the Amazon-Photo, Amazon-Computers, and Minesweeper datasets. Most notably, BDFormer achieves best performance ($97.01\%$) on Amazon-Photo, surpassing Polynormer's reported peak of $96.46\%$. These results indicate that the performance gains of BDFormer remain consistent under larger model capacities.

*Table 7.* Performance comparison under the setup of Polynormer.

| Model | Photo | Computers | Minesweeper |
|---|---|---|---|
| Polynormer | $96.46_{\pm 0.26}$ | $93.68_{\pm 0.21}$ | $97.46_{\pm 0.36}$ |
| BDFormer | $97.01_{\pm 0.20}$ | $93.82_{\pm 0.25}$ | $97.55_{\pm 0.31}$ |

## C.5. Performance in Unsupervised Initialization Settings

As discussed, the default BDFormer leverages class centroids to provide a warm start for the latent anchors. To investigate the model's dependency on this class-guided prior, an extreme unsupervised initialization scenario is evaluated. Specifically, the class-guided warm start is replaced with a standard K-means applied to the node attributes $\mathbf{X}$. The resulting $k$ centroids are utilized as the initial states for $\mathbf{C}$, while the rest of the semi-supervised training pipeline remains unchanged.

**Results and Analysis.** Tab. 8 reports the performance under this unsupervised initialization setting. As expected, lacking a label-aware warm start incurs a slight performance degradation. However, the accuracy gap is marginal (*e.g.*, less than a $1\%$ drop on Cora, CiteSeer, and Squirrel) compared to the default model. This demonstrates that the model preserves strong discriminative power without relying on initial label distributions. This robustness can be attributed to the spectral block regularization $\mathcal{L}_{spec}$. By discovering and enforcing the latent block-diagonal structure, $\mathcal{L}_{spec}$ prevents semantic collapse and guides the propagation process to a highly competitive state, even when starting from a feature-based K-means initialization.

*Table 8.* Impact of unsupervised initialization on model performance.

| Setting / Initialization | Cora | CiteSeer | Chameleon | Squirrel |
|---|---|---|---|---|
| Semi-supervised (Default ) | 89.01 | 77.53 | 47.90 | 44.20 |
| Unsupervised (K-means) | 88.72 (0.29 ↓) | 76.98 (0.55 ↓) | 46.39 (1.51 ↓) | 43.67 (0.53 ↓) |

