# OpenReview forum: "Rethinking Graph Transformers as Graph Signal Denoisers: The Role of Block-Diagonal Priors"
_ICML.cc/2026/Conference — ICML 2026 regular_

### Official Review · Reviewer_GEga · 2026-03-08

**Soundness:** 3
**Presentation:** 4
**Significance:** 3
**Originality:** 4
**Overall Recommendation:** 5
**Confidence:** 5

**Summary:**

The paper investigates a theoretical unification of graph transformer (GT) designs through the lens of graph signal denoising (GSP). The authors demonstrate that a block-diagonal structure in the propagation operator is a key inductive bias for enhancing representation quality. Based on this, they propose BDFORMER, which utilizes spectral-regularized cross-attention on latent anchors to enforce this structural prior while maintaining linear complexity. Empirical results show competitive performance of BDFORMER across various benchmarks.

**Compliance With Llm Reviewing Policy:**

Affirmed.

**Final Justification:**

The authors have successfully addressed all of my concerns. I have raised my score by one point.

**Key Questions For Authors:**

(i)In heterophilic datasets, how do you ensure the learned blocks capture structural equivalence rather than just forcing label-based clustering?

 (ii) What specific numerical safeguards (e.g., epsilon-perturbation) are implemented to ensure stable backpropagation through the spectral loss layer?

(iii) Does enforcing a strict block-diagonal structure increase the risk of over-smoothing within blocks? How does the model maintain fine-grained node-level information in large clusters?

(iv) Is the local topology refinement dynamic throughout training? Can the model recover a pruned edge if the global anchor affinity shifts in later epochs?

**Limitations:**

yes

**Strengths And Weaknesses:**

**Strengths:**

(i)The idea of a block-diagonal prior is intriguing and valuable.

(ii) The proposed GT is well-motivated, simple, and effective.

(iii) The paper is well-organized, maintaining a good progression from theoretical derivation to its practical implementation.


**Weaknesses:**

(i)The block-diagonal prior is rooted in homophily, yet the paper lacks discussion on its impact on structural heterophily.

(ii) Using eigenvalue decomposition during training poses potential gradient instability (e.g., exploding/vanishing gradients).

---

> ### Author Rebuttal · Authors · 2026-03-31
>
> We sincerely thank you for recognizing our work. We address your questions below.
>
> > W1&Q1: Heterophily and Structural Equivalence
>
> R1. In heterophilic graphs, our block-diagonal prior captures structural equivalence rather than merely forcing label-based clustering. This is guaranteed by both our algorithmic design and empirical evidence:
>
> - While partial labels provide a "warm start" for anchor initialization, they do not dictate the final structure. During training, the anchors are continuously updated by aggregating projected features from all nodes in the graph (both labeled and unlabeled). This ensures that the anchors evolve to capture the global data distribution and latent contexts, rather than being strictly bounded by initial labels. In heterophily, structurally equivalent nodes (e.g., nodes of the same class that are disconnected but share similar attributes or roles) will naturally exhibit similar affinity distributions toward these globally updated anchors. Therefore, they are grouped into the same spectral block.
>
> - To definitively rule out “label-forcing”, we have added a new experiment, as detailed in our response to Reviewer CkKv (R1). When we replace the label-guided initialization with completely unsupervised k-means clustering, the model still achieves highly competitive performance on heterophilic graphs. It incurs only a small drop on Chameleon and Squirrel, while still outperforming most baseline models. This empirically proves that the learned blocks intrinsically capture structural equivalence independent of label guidance.
>
> > W2&Q2: Numerical Safeguards for Spectral Loss
>
> R2. It is absolutely correct that backpropagating through the eigen-decomposition requires strict numerical handling to prevent gradient instability (e.g., NaNs or exploding gradients). In practice, we implemented a comprehensive set of safeguards in the spectral loss layer:
>
> - We add a small perturbation ($\epsilon * I$, with $\epsilon=10^{-6}$ cast in float64 for numerical precision) to the affinity matrix before computing the Laplacian. This breaks exact singularities and stabilizes the underlying torch.linalg.eigh solver during the backward pass.
>
> - When computing the normalized Laplacian ($L=I−D^{−\frac{1}{2}}AD^{−\frac{1}{2}}$), we explicitly clamp the row-sums (node degrees) with a minimum value of $1e^{-6}$. This strictly prevents division-by-zero or inf/NaN gradients during the $rsqrt()$ operation.
>
> - To prevent floating-point inaccuracies from violating the symmetric property required by eigh, we explicitly enforce symmetry via $A^{'}=\frac{1}{2}(A+A^{\top})$ and clamp the affinity values to be non-negative.
>
> These combined operations provide robust mathematical safeguards that ensure bounded gradients and stable convergence across all our experiments.
>
> > Q3: Over-smoothing and Fine-grained Node Information
>
> R3. Enforcing a block-diagonal structure inherently mitigates harmful over-smoothing rather than exacerbating it. We maintain fine-grained information through two theoretical and architectural mechanisms:
>
> - Intra-block Smoothing vs. Global Over-smoothing: Traditional over-smoothing degrades performance because it homogenizes features across the entire graph, causing nodes of different classes to become indistinguishable (inter-class collapse). In contrast, our block group has structurally equivalent nodes. Smoothing within a block acts as a beneficial feature denoising (increasing intra-class compactness). Crucially, the block-diagonal prior strictly prevents attention from leaking across different blocks, thereby halting global over-smoothing.
>
> - Within a large cluster, fine-grained node-level identity is never lost. This is preserved by the residual connections ($H^{+}=H+H_{Attn}$). The node's original, unique feature acts as the dominant base, while the anchor-propagated context serves only as an additive structural refinement. Thus, nodes in the same block share a macroscopic context but do not collapse into identical representations.
>
> > Q4: Dynamic Local Topology Refinement
>
> R4. Yes, the local topology refinement is fully dynamic. The global anchor affinity and the resulting confidence mask are re-evaluated continuously during forward propagation. Because the underlying adjacency structure is preserved, if the model updates its anchor representations and shifts global affinities in later epochs, a "pruned" edge from previous epochs can be fully recovered and utilized in the subsequent epochs. This dynamic property is highly beneficial, as it allows the model to correct early pruning mistakes as the representations mature.

---

> > ### Author Rebuttal · Reviewer_GEga · 2026-04-01
> >
> > Thanks for your rebuttal, the authors have addressed my major concerns. I maintain my recommendation.

---

### Official Review · Reviewer_CkKv · 2026-03-08

**Soundness:** 2
**Presentation:** 2
**Significance:** 1
**Originality:** 2
**Overall Recommendation:** 4
**Confidence:** 4

**Summary:**

The paper introduces BDFORMER, a Graph Transformer (GT) architecture designed to improve node classification by enforcing a block-diagonal structural prior on the representation propagation operator. The authors unify existing GT designs under a Graph-Signal Denoising (GSD) framework, proving that denoising efficacy is fundamentally dictated by the block-diagonal structure of the propagation matrix.

To enforce this structure, the model introduces latent anchors and applies spectral regularization to encourage a block-diagonal anchor affinity matrix. The learned global structure is also used to prune heterophilous edges in the local graph topology.

**Compliance With Llm Reviewing Policy:**

Affirmed.

**Final Justification:**

I thank the authors for their additional experiments. Their new results have addressed all my concerns, and I believe have largely strengthened several scalability claims made in the paper. Therefore, I will increase my score to weak accept.

**Key Questions For Authors:**

- Figure 5 shows that performance gains marginalize after $k \ge 64$ and $g \ge 10$ on relatively small graphs. For larger graphs like ogbn-proteins, do the optimal $k$ and $g$ need to scale with the number of nodes, or is a small, fixed number of anchors sufficient for any graph size?
- The paper uses a trade-off parameter $\lambda$ to fuse local and global representations. Can the authors provide more insight into the optimal $\lambda$ values across different datasets? Is the local channel more critical for homophilic graphs than for heterophilic ones?
- How sensitive is the training stability to the discretization of attention scores via STE?

**Limitations:**

The paper does not discuss the limitations. For example:
Much of the methodology, particularly the anchor initialization using class centroids from labeled training nodes, is tailored for transductive semi-supervised node classification. Its applicability to fully inductive tasks or graphs with zero initial labels remains a limitation.

I think a separate subsection should be used to discuss limitations and potential caveats more systematically.

**Strengths And Weaknesses:**

### Strengths
- Clear motivation and an interesting theoretical perspective based on graph signal denoising.
- The idea of imposing a block-diagonal structural prior through anchor-based attention is novel and potentially useful.

### Weaknesses
- While the paper proposes a label-guided orthogonal initialization for anchors, the model's performance in strictly unsupervised or inductive settings where high-quality class centroids might not be available is not fully explored. This would largely limit the applicability of the proposed method in unsupervised or self-supervised settings.
- The model introduces several new hyperparameters, including the number of anchors ($k$), the number of blocks ($g$), the spectral coefficient ($\beta$), the edge preservation ratio ($p$), and the trade-off parameter ($\lambda$). Managing this increased search space may complicate deployment compared to simpler GT baselines, thereby making the comparison less fair.
- The performance numbers of some baselines differ from those reported in the original paper. For instance, for Polynormer, this paper reported 95.44 vs 96.46 in the original paper on Photo. It is not clear where this difference comes from.
- Several claims about the scalability were made, but the largest graphs considered have only ~100k nodes. Evaluating on ogbn-products is necessary to understand whether the proposed method generalizes well on graphs with millions of nodes.
- Although the overall complexity is linear, the spectral regularization requires an eigen-decomposition of the $k \times k$ anchor affinity matrix ($O(k^3)$). While $k$ is small, this still represents a potential bottleneck if $k$ were to increase significantly for more complex datasets. Figure 5 only presents its impact for small graphs.

---

> ### Author Rebuttal · Authors · 2026-03-31
>
> We deeply thank you for the constructive feedback. We address your concerns below.
>
> > W1&Limitations: Initialization Strategy
>
> R1. We agree that exploring strictly unsupervised settings is crucial, particularly when high-quality class centroids are unavailable. To address this, we adapted BDFormer to a fully unsupervised framework (0% labels) by replacing the label-guided initialization with simple K-means clustering.
>
> As shown in **[Table](https://anonymous.4open.science/r/BDFormer-demo/exp_ini_loss.pdf)**, BDFormer (K-means) maintains highly competitive performance. This success is not coincidental; it is fundamentally driven by our Spectral Block Regularization ($L_{spec}$). When high-quality semantic centroids are absent, $L_{spec}$ effectively compensates by enforcing a strong structural prior on the latent anchors, guiding the model to discover intrinsic topological communities. To explicitly prove this, we removed $L_{spec}$ under the unsupervised setting. The resulting significant performance decline confirms that $L_{spec}$ is the critical driving force for unsupervised applicability.
>
> We will add a Limitations subsection to discuss these applicability boundaries.
>
> > W2: Hyperparameter Search Space and Tuning Burden
>
> R2. The introduced parameters do not complicate deployment or compromise fair comparison. The search space is highly compact:
> - Zero Tuning: The pruning ratio $\rho$ is permanently fixed at 0.8.
> - Stable Performance (Figs. 5 and 8): Parameters $k$, $g$, and $\beta$ exhibit remarkable stability. Performance predictably plateaus once they reach moderate thresholds, avoiding the need for fine-grained search.
> - Minimal Tuning: Only the local-global trade-off $\lambda$ requires standard dataset-specific tuning.
>
> Therefore, introducing these parameters does not complicate deployment compared to baseline models.
>
> > W3: Discrepancy in Performance (Polynormer)
>
> R3: The reported 95.44 for Polynormer stems from our unified evaluation protocol (as detailed in Appendix B.2.2) designed to ensure fair comparison under identical resource constraints.
>
> All models (including ours) were tuned via grid search over a unified hyperparameter space: hidden dimension in $\\{64,128,256\\}$, number of layers in $\\{1,2,3,4\\}$, and attention heads fixed at 4; Only for parameters unique to each baseline (e.g., warm-up epochs in Polynormer) did we adopt the values recommended in their papers (Lines 800-801). The peak score of 96.46 reported in the Polynormer paper was achieved using a massive architecture (9 layers, 512 hidden dimensions, 8 heads). When evaluated under our unified fair-comparison space, its capacity naturally adjusts, yielding 95.44.
>
> Under these same resource constraints, BDFormer achieves 96.27 on Photo, outperforming the constrained Polynormer and nearly matching its unconstrained peak. This highlights the superior parameter efficiency of our architecture.
>
> > W4, W5&Q1: Scalability on Million-scale Graphs and $O(k^3)$ Complexity
>
> R4. The concern regarding the $O(k^3)$ bottleneck logically assumes that $k$ (anchors) and $g$ (blocks) must scale proportionally with $N$ (nodes). We clarify that $k$ and $g$ capture the graph's intrinsic latent communities, which depend on the distribution complexity, not the node count. A larger graph represents denser sampling, not a proportional explosion of structural modes. Thus, $k$ remains tightly bounded. As shown in the ogbn-proteins ablation (see **[Table](https://anonymous.4open.science/r/BDFormer-demo/exp_k.pdf)** and **[Table](https://anonymous.4open.science/r/BDFormer-demo/exp_g.pdf)**), performance saturates at $k=128$ and $g=10$.
>
> Because $k$ is small and bounded ($k\le256$), the $O(k^3)$ eigen-decomposition operates on a tiny $256 \times 256$ matrix. It acts as a negligible $O(1)$ overhead, dwarfed by the $O(N)$ feature transformations.
> To prove this, we evaluated BDFormer on the 2.4 million-node ogbn-products dataset (see **[Table](https://anonymous.4open.science/r/BDFormer-demo/exp_per_time_gpu.pdf)**). With only $k=256$, BDFormer achieves superior accuracy and maintains highly efficient training, confirming its million-scale applicability.
>
> >Q2&Q3: Mechanism Insights ($\lambda$ and STE Stability)
>
> Optimal $\lambda$: Empirically, the optimal $\lambda$ does not strictly correlate with the overall homophily ratio. While small graphs align with intuition (e.g., homophilic CiteSeer prefers local channels at 0.3; heterophilic Chameleon prefers global at 0.8), this dichotomy fails on larger graphs. Large-scale topologies often introduce competing structural factors, such as long-range dependencies, that go beyond simple homophily-based rules. Therefore, $\lambda$ is designed as a dataset-specific mechanism.
>
> STE Stability: Despite the gradient mismatch of STE, training is highly stable because $L_{spec}$ regularizes the continuous attention logits before hard discretization. This provides a smooth structural prior that prevents erratic gradient updates.

---

> > ### Author Rebuttal · Reviewer_CkKv · 2026-04-03
> >
> > I thank the authors for their detailed rebuttal. Please find below my remaining questions and concerns.
> >
> > > W2: Hyperparameter Search Space and Tuning Burden
> >
> > To make the comparison more direct, could the authors report **runtime comparisons for all methods in Table 1 on a few datasets** (e.g., Cora, Photo, and Squirrel), **including the time spent on hyperparameter search**? I believe such results would strengthen the claim that the proposed method does not introduce additional tuning burden.
> >
> > > W3: Discrepancy in Performance (Polynormer)
> >
> > Thank you for the clarification. However, to make the comparison fully convincing, it would be helpful if the authors could provide **a complete comparison under the same resource constraints used for Polynormer**. I believe that such a controlled comparison would better support the state-of-the-art performance claims made in the paper.
> >
> > > Scalability on Million-scale Graphs
> >
> > Standard deviations are not reported for the results on ogbn-products. Were these numbers obtained from a single run, or averaged over multiple runs?

---

> > > ### Author Response · Authors · 2026-04-06
> > >
> > > We sincerely thank you for your valuable time. Below is our response to your concerns.
> > >
> > > > W2: Tuning Burden & Runtime
> > >
> > > R1: Following your advice, we reran all methods and reported the total training time (including hyperparameter search) for the Cora, Photo, and Squirrel, consistent with our established protocol. The experiments were conducted using parallel RTX 3090 GPUs.
> > >
> > > As shown in **[Table A](https://anonymous.4open.science/r/BDFormer-demo/exp_tuning_time.pdf)** and **[Table B](https://anonymous.4open.science/r/BDFormer-demo/exp_variant.pdf)**, the results directly substantiate that BDFormer achieves superior accuracy while maintaining a practical tuning profile, proving that our model does not introduce additional obligatory tuning burden:
> > > - Significantly faster than heavy Transformers (Table A): Exphormer and GraphGPS approximately demand up to 220h and 384h on the Photo dataset. In contrast, BDFormer's strict linear complexity guarantees exceptionally fast runtimes, fully offsetting its structural grid and capping the total exhaustive search time at 62h—an order of magnitude lower.
> > > - Manageable burden on par with advanced baselines (Table A): While our standard grid naturally requires more tuning than ultra-lightweight linear models (e.g., NodeFormer), **BDFormer's total cost aligns closely with other high-performance models (e.g., PolyNormer, GloGNN) and remains completely manageable**.
> > > - Minimal obligatory tuning burden in practice (Table B): To address the concern regarding tuning burden, we evaluated a BDFormer variant denoted BDFormer$^{\star}$. In this variant, we completely fix the parameters ($g=15, \beta=0.01$) as universal defaults across all datasets. Remarkably, this drastically reduces total tuning time to **<7.5h** (Table A), yet **BDFormer$^{\star}$ remains highly competitive with the fully-tuned SOTA baseline M$^{3}$Dphormer**—outperforming or matching it on 9 out of 12 datasets.
> > >
> > > In short, the exhaustive grid search is merely an option to extract the theoretical upper bound. In practical deployments, BDFormer serves as a robust architecture without the heavy tuning burden.
> > >
> > > > W3: Comparison with Polynormer
> > >
> > > R2: Following your guidance, we evaluated BDFormer and Polynormer on Photo, Computers, and Minesweeper under the setup used in Polynormer. Specifically, we strictly aligned the model capacity by adopting their configuration ranges: 512 hidden dimension, 8 attention heads, dropout $\in \\{0.2, 0.3, 0.5, 0.7\\}$, and local/global layers $\in \\{1, 2, ..., 7\\}$.
> > >
> > > As shown in **[Table](https://anonymous.4open.science/r/BDFormer-demo/exp_polynormer.pdf)**, when evaluated under these identical resource constraints, BDFormer consistently outperforms Polynormer. We will add these supplementary results and the configuration details to the revision.
> > >
> > >
> > > > Standard Deviations
> > >
> > > R3: The results reported in the previous response were the average of 5 independent runs; however, we omitted the standard deviations when formatting the tables. We have corrected this oversight (see **[Table](https://anonymous.4open.science/r/BDFormer-demo/exp_per_time_gpu.pdf)**).

---

### Official Review · Reviewer_Qgi8 · 2026-03-11

**Soundness:** 3
**Presentation:** 3
**Significance:** 3
**Originality:** 3
**Overall Recommendation:** 4
**Confidence:** 3

**Summary:**

This paper studies graph transformers for node classification through a unifying message-passing perspective. It argues that both score-level and representation-level topology injection can be viewed as graph signal denoising, and that a block-diagonal propagation operator is the desired structure for filtering heterophilous noise while preserving intra-class propagation. Based on this view, the paper proposes BDFormer, a dual-channel architecture that combines anchor-based global cross-attention, spectral regularization on anchor affinities, and local topology pruning guided by learned global affinities. Experiments on homophilic, heterophilic, large-scale, and long-range benchmarks report competitive or best performance against a broad set of GNN and graph transformer baselines.

**Compliance With Llm Reviewing Policy:**

Affirmed.

**Final Justification:**

All my concerns were addressed.

**Key Questions For Authors:**

Refer weaknesses

**Limitations:**

Yes

**Strengths And Weaknesses:**

**Strengths:**

1. The paper has a clear high-level motivation. Figure 1 does a good job explaining the core intuition, namely that both the observed graph and unconstrained global attention can inject noise, and that the target is a more block-structured propagation pattern.
2. The empirical scope is fairly broad. Table 1, Table 2, and Table 3 cover homophilic, heterophilic, large-scale, and long-range settings, and the baseline set is stronger.
3. The qualitative visualizations are helpful. In Figure 2, BDFormer does appear to learn more structured attention patterns than GT, NodeFormer, and SGFormer, which at least supports the intended mechanism qualitatively.
4. The approach is practically motivated. Using anchors to factorize global interactions is a sensible route to linear-time graph transformers, and Figure 4 is at least consistent with the claimed scaling trend.

**Weaknesses:**

1. The paper leans heavily on Theorem 1 and the accompanying text in Section 3.1 to claim an equivalence between MP and GSD. But the appendix proof does not establish an exact equivalence for the stated objective in Equation 6 when $ \mathbf{P} $ is non-symmetric. Instead, it swaps in an approximate gradient direction. That is a much weaker statement. If the theory is meant as intuition, present it as intuition. If it is meant as theorem-level support, it needs to be correct as written.
2. The paper repeatedly says the ideal propagation operator is block-diagonal and that this structure is the theoretical optimum for denoising. I could not find a theorem in the main paper that proves such an optimality result under explicit assumptions. Homophily-based intuition is fine, but an intuition does not justify the language used in the introduction and methodology.
3. Method specification is incomplete in important places. The transform $ \psi(\cdot) $ in Equation 17 is undefined in the main paper. The phrase “label-guided orthogonal initialization” does not match Equation 8, which does not impose orthogonality. Also, the relation among the number of anchors (k), the number of blocks (g), and the number of classes (c) is underexplained.
4. Scalability is argued more than demonstrated in the main paper. Figure 4 only shows that BDFormer scales roughly linearly with input size on sampled ogbn-arxiv subgraphs. It does not compare against dense-attention transformers, and it does not establish the claimed efficiency advantage relative to other linear GTs under matched conditions. The appendix helps, but some pieces should be in the main paper.
5. Several typos and inconsistencies in the paper.

---

> ### Author Rebuttal · Authors · 2026-03-31
>
> We deeply thank you for the positive feedback. We address your concerns below.
>
> > W1: Equivalence under Non-symmetric $P$
>
> R1. You are correct that for a non-symmetric propagation operator $P$ (such as Softmax attention), the relationship established in Appendix A.1 represents a first-order gradient approximation rather than a strict symmetric equivalence. In the context of graph neural diffusion [1] and unified optimization framework [2], applying an asymmetric, row-stochastic attention matrix is well-recognized as a first-order numerical realization of the underlying Dirichlet energy functional.
>
> We will revise the manuscript and appendix to explicitly characterize our formulation as a principled gradient-based approximation of GSD, rather than a strict equivalence.
>
> [1] GRAND: Graph Neural Diffusion, ICML’21
>
> [2] Interpreting and Unifying Graph Neural Networks with An Optimization Framework, WWW’21
>
> > W2: Formalizing the "Optimality" of the Block-Diagonal Structure
>
> R2. We respectfully clarify that characterizing the block-diagonal structure as the "theoretical optimum" is **not** merely homophily-based intuition, but a direct mathematical consequence of minimizing the Graph-Signal Denoising (GSD) objective (Eq. 6) within our framework.
>
> The GSD objective explicitly penalizes feature differences between connected nodes. In a graph containing distinct semantic manifolds, inter-class edges inevitably connect nodes with disparate features. Therefore, any propagation operator with non-zero inter-class transition probabilities will strictly increase the high-frequency noise penalty in Eq. 6. To globally minimize Eq. 6—achieving maximum intra-class smoothing without aggregating inter-class noise—the propagation matrix must structurally isolate distinct manifolds. Mathematically, this zero-interference condition dictates a strictly block-diagonal matrix. By contrast, the dense, unconstrained attention matrices used in standard GTs inherently violate this condition, leading to suboptimal GSD minimization.
>
> We acknowledge the point that this analytical derivation was presented descriptively rather than as a formal theorem. In the revision, we will upgrade this discussion into a formal Proposition, explicitly stating the assumptions (i.e., minimizing Eq. 6 over disjoint semantic manifolds) under which the block-diagonal structure is the unique analytical minimizer.
>
> > W3: Methodological Specifications ($\psi()$ and $k$,$g$,$c$)
>
> R3. We apologize for the incomplete specifications in the main text. We will clarify these operational details in the revision:
>
> Transform $\psi()$: this is a necessary post-processing operator applied to the raw anchor inner-product matrix ($BB^{\top}$). It ensures the matrix is symmetric, non-negative, and numerically stable before degree normalization by performing symmetrization, a ReLU activation, and adding a small diagonal jitter: $A_B=max(\frac{1}{2}(A_c+A_c^{\top}),0)+\epsilon I$, where $A_c=BB^{\top}$, where $\epsilon=10^{-6}$.
>
> We acknowledge that the phrase "label-guided orthogonal initialization" in the text contradicts Eq. 8. Eq. 8 reflects our true implementation: independent Gaussian noise is added to anchor features obtained from MLP-projected class centroids. We will correct the text to "class-guided anchor initialization" to ensure consistency.
>
> Relation among $k$,$g$, and $c$: These parameters form a conceptual structural hierarchy:
> - $k$ (Micro-level Anchors): Provides an over-complete basis to capture the multimodal feature distributions of each class (For a detailed theoretical justification on why setting $k>c$, please refer to our response R3 to Reviewer BUd9).
> - $g$ (Meso-level Blocks): Acts as a structural bottleneck where the $k$ micro-anchors are grouped into $g$ communities via spectral loss.
> - $c$ (Macro-level Semantics): The final number of ground-truth classes.
>
> > W4: Scalability and Baselines Comparison
>
> R4. We will move the efficiency analysis (Appendix B.3.2, Fig. 6) to Section 4.2.
>
> To address the scalability concerns, we clarify our baselines and provide new large-scale evidence. The purpose of Fig. 4 is strictly to validate the linear $O(N)$ scaling behavior of BDFormer, rather than to serve as a comparative benchmark. Standard dense GTs (e.g., SAT [1]) incur an $O(N^2)$ complexity, which directly causes OOM errors on even the smallest graphs in this experiment (starting from 20K nodes).
>
> To demonstrate scalability beyond ogbn-arxiv (Fig. 6), we evaluated BDFORMER on the massive ogbn-products dataset (∼2.4 million nodes), as shown in **[Table](https://anonymous.4open.science/r/BDFormer-demo/exp_per_time_gpu.pdf)**. This confirms that our theoretical linear complexity translates directly to practical large-scale scalability.
>
> [1] Structure-aware transformer for graph representation learning, ICML’22
>
> > W5: Typos and Inconsistencies
>
> R5. We will carefully proofread the manuscript to correct typos and ensure consistent notation and terminology.

---

> > ### Author Rebuttal · Reviewer_Qgi8 · 2026-04-03
> >
> > Thanks for the detailed response. All my primary concerns have been addressed and hence I have increased the score by one point.

---

### Official Review · Reviewer_BUd9 · 2026-03-15

**Soundness:** 3
**Presentation:** 3
**Significance:** 2
**Originality:** 2
**Overall Recommendation:** 4
**Confidence:** 4

**Summary:**

The authors unify graph transformers (GT) that induce topology through (i) score-level and representation-level topology injections through the lens of message-passing and graph signal denoising. Based on this unification they propose BDFormer, a GT that leverages semantically aligned “anchors” and dual-channel propagation between anchors and nodes to arrive at denoised node-level representations while avoiding quadratic-runtime limitations of standard GTs. The authors demonstrate the improved performance and scalability of BDFormer through a series of experiments and provide a mechanism analysis.

**Compliance With Llm Reviewing Policy:**

Affirmed.

**Final Justification:**

I thank the authors for the further explanations re: W2, with all other concerns already largely addressed in the initial rebuttal. The provided additional analysis for W2, while largely mechanistic and post-hoc in nature, is indeed helpful in understanding how BDFormer may leverage these latent semantic clusters, so I hope to see these points added to the revision. I think providing a more in-depth analysis with perhaps additional theoretical grounding on this end would serve as valuable future work.

I commend the authors for a successful rebuttal, and am happy to increase my score by one point.

**Key Questions For Authors:**

1. Following Weakness 3:
   1. Is there an inherent semantic meaning to the number of anchors $k$, or its role in settings when set to $k=c$ (i.e., no redundancy) or $k >> c$? If BDFormer is sensitive to $k$, why is that so?
   2. Same questions, but applied to $\rho$.
2. To what extent is the block-diagonal structure maintained, since the spectral constraint is applied to only the initial structure $\mathbf{B}^{(0)}$? Would there be an additional benefit to more strongly enforcing it beyond initialization?
3. I had some trouble validating the following claim (lines 405-408), can you add some clarification? — if anything, to me the pruned graph in Fig. 3 seems to put more weight on the off-diagonal:
> Conversely, the heatmaps of the pruned graph demonstrate an slight improvement: edge weights are redistributed along the main diagonal, suppressing offdiagonal interference compared to the input.

**Limitations:**

The authors have not explicitly discussed the limitations underlined in the Weaknesses section.

Despite the convincing overall design and empirical performance of the BDFormer model, I think this performance is highly dependent on high label availability, which may hamper its applicability beyond benchmark datasets. This, and the potential gaps in the theoretical motivation of the paper lead me to recommending a weak reject in its current state.

**Strengths And Weaknesses:**

**Strengths:**
1. I like the overall narrative and flow of the paper; while this is a mostly method-driven paper in my opinion, the authors provide some theoretical backing behind the design decisions made, and while perhaps the contributions aren’t groundbreaking and rather are in the form of small individual contributions, the resulting work is cohesive and seemingly effective.
2. The experimental section is well-structured and comprehensive, with a clear narrative in mind. I appreciate that the authors go beyond pure performance and address scalability, long-range dependencies, heterophilic settings as well as an extensive mechanism analysis. I also appreciate the large set of benchmarks compared against.

**Weaknesses:**
1. The initial anchor representations heavily rely on access to class labels. I believe all datasets except Paris & Shanghai have 50%+ labeled nodes, making the initial centroids very reliable. Paris & Shanghai have 10% labeled nodes, but the large graphs still mean that each dataset has access to 10K+ labeled nodes for classification over 10 classes. I think this step almost trivializes the block-diagonal construction in the settings BDFormer is evaluated on. An ablation study evaluating BDFormer reliance on this step (e.g. testing with random initialization as well as increasing label percentage from 1% to 50% on multiple datasets, and at least one heterophilic one) is necessary.
2. The theoretical motivation feels somewhat superficial to me upon close inpection. The motivation mostly stems from the point that nodes should message-pass within their class neighborhoods (rather than topological closeness which may inject noise): My understanding is that BDFormer is particularly effective in the heterophilic datasets as it gathers same-class nodes into the same blocks in the diagonal so that they can message-pass more effectively and ignore topological noise coming from the graph structure itself. The authors denote that $\mathbf{P}$ should be block-diagonal with respect to the global structure, and then they proceed to define the global structure over the _class labels_ (as per my previous point). I think the supposed theoretical contribution here is somewhat tautological, in the sense that aligning the block-diagonal with the “global structure” is of course optimal if the global structure is defined by the class labels.
3. There is no principle for selecting (i) for the number of anchors $k$, or (ii) the pruning ratio $\rho$. To me these seem to be an empirically-driven heuristics that are not motivated theoretically (or any other way, for that matter).
   - $k$ is determined per dataset by grid search over $\{32, 64, 128, 256\}$, though as per Fig. 5 this seems to have minimal effect on performance. I suspect $k > c$ (# classes) for all/most datasets, so many classes end up being assigned to multiple anchors. Why not just set $k = c$, for example? The inherent meaning of these anchors (in particular the redundant ones) are not clear to me, and the authors provide no intuition on this.
   - Similarly $\rho$ is fixed to 0.8 for all experiments, with no justification whatsoever.  At least a sensitivity analysis of $\rho$ would be welcome.
4. Minor issues & typos (no effect on score):
   1. Line 254-255: Malformed sentence: “..., he global attention patterns to adaptively is leveraged to refine the graph topology.”
   2. The proof of Theorem 2 isn’t referenced in the main body.
   3. Some figures (e.g. 4 and 5) should mention the datasets the plots are based on in the captions.

---

> ### Author Rebuttal · Authors · 2026-03-31
>
> We sincerely thank you for the kind recognition of our work. We address your concerns below.
>
> >W1: Reliance on Labels for Anchor Initialization
>
> R1. To address the concern regarding label reliance, we ablated the initial label percentages (50%, 30%, 10%, 1%) and evaluated completely unsupervised initializations (Random and K-means). Results are presented in **[Table](https://anonymous.4open.science/r/BDFormer-demo/exp_labels.pdf)**.
>
> - BDFormer does **not** rely on massive labeled sets. With merely 1% labels, it remains highly competitive. This confirms the model effectively learns structural representations rather than simply memorizing initial dense labels.
> - Removing label initialization and using K-means clustering yields strong results that nearly match the 10%-30% label setup. While pure random initialization naturally degrades performance as it strips the model of any structural prior, the empirical success of K-means definitively proves that BDFormer captures global structural distributions.
>
> > W2: Theoretical Motivation and "Tautology" Concern
>
> R2. The theoretical motivation is **not** tautological, because our motivation is rooted in GSP, independent of supervised labels.
>
> - Theoretical Optimality (Theorem 1): A block-diagonal operator $P$ acts as an optimal low-pass filter for signal denoising. It minimizes Dirichlet energy within latent communities and cuts off inter-community noise. This mathematical optimality holds regardless of downstream labels.
>
> - Latent Structure $\neq$ Labels: BDFORMER captures latent structural equivalence. In complex graphs, a single class often exhibits a multimodal distribution spanning multiple latent clusters. Labels solely provide a "warm start"; the learned block-diagonal structure models these intrinsic topological communities, not a 1-to-1 label mapping.
>
> - The strong performance of our fully unsupervised variant (0% labels via K-means, see W1) explicitly confirms that the block-diagonal structure captures intrinsic topological features rather than being a tautology of labels, fully validating our theoretical motivation.
>
> **Summary**: We believe the new settings—extreme label scarcity (1%) and fully unsupervised initialization—definitively close these gaps. Together, they prove that BDFormer does not rely on dense annotations for real-world applicability.
>
> We are grateful to you for guiding us. As suggested, we will add these discussion into Limitations section.
> > W3 & Q1: Intuition for $k$ and $\rho$
>
> R3. These are principled parameters governing representation capacity and noise-filtering:
>
> Intuition for $k$: Setting $k=c$ erroneously assumes each class follows a simple unimodal distribution. In complex graphs, a semantic class typically forms a multimodal distribution. Setting $k>c$ provides an over-complete basis (following [1]), allowing micro-anchors to independently capture the multimodal, structural sub-clusters within a single semantic class.
>
> Pruning Ratio ($\rho$): $\rho$ acts as a structural noise filter. A high threshold ($\rho=0.8$) prunes spurious interactions while keeping the essential structural backbone intact. Sensitivity analysis (in **[Table](https://anonymous.4open.science/r/BDFormer-demo/exp_sens_rho.pdf)**) shows that performance is robust across $\rho \in \\{0.6, 0.8\\}$, degrading at essential structural backbone are severed ($\rho \le 0.5$) or extremes where noise dominates ($\rho = 1$).
>
> [1] Deep Clustering for Unsupervised Learning of Visual Features, ECCV’18
>
> > Q2: Maintenance of Block-Diagonal Structure and Deep Constraints
>
> R4. The global structure does not decay in deeper layers because $B^{(l)}$ depends directly on $B^{(0)}$ (Eq. 10) instead of $B^{(l−1)}$. This explicit architectural injection guarantees that the global block-diagonal prior established at initialization persistently anchors the structural semantics.
>
> Enforcing the spectral constraint on every layer (BDFormer-All) yields no additional benefit and slightly degrades performance (see **[Table](https://anonymous.4open.science/r/BDFormer-demo/exp_abl_layer.pdf)**). This may be attributed to the fact that enforcing constraints beyond initialization causes over-regularization, restricting functional plasticity in deeper layers.
>
> > Q3: Clarification on Figure 3
>
> R5. The perceived prominence of some off-diagonal blocks is a visual artifact caused by the increased sparsity (which makes the remaining edges stand out against the white background). A block-by-block inspection confirms off-diagonal suppression and diagonal enhancement:
> - Chameleon: Off-diagonal blocks ((0,1), (1,0), (3,4), (4,3)) become visibly lighter. And, diagonal blocks ((1,1), (3,3)) significantly deepen in color.
> - CiteSeer: A similar enhancement is observed, with diagonal blocks ((2,2), (3,3), (4,4), (5,5)) intensifying.
>
> We will add these specific block-wise comparisons to the revision to prevent misinterpretation.
>
> > Minor issues & typos:
>
> We will polish the manuscript and fix all minor issues.

---

> > ### Author Rebuttal · Reviewer_BUd9 · 2026-04-04
> >
> > - **[W1]** I appreciate the convincing additional study -- this is addressed.
> > - **[W3]** The intuition for $k$ is sound. The argument for $\rho$ isn’t necessarily “principled”, it is determined empirically and may perhaps vary per dataset (this is only a minor concern).
> > - **[Q2, Q3]** Acknowledged, thanks.
> >
> > I think this is largely a good rebuttal, but I would like to follow-up regarding **W2**: Firstly, I do concede that when paired with W1, the claims are not tautological, and they rely on “latent semantic clusters” and not necessarily class _labels_. However, while arguing for the mechanism at play relies on latent semantic clusters (which the block-diagonal $\mathbf{P}$ should correspond to) is reasonable, I don’t think it’s very convincingly demonstrated.
> > - For one, the authors leverage class centroids in Eq. 8, which essentially concedes that class labels do provide a good initialization (so latent structure $\neq$ labels, but they’re typically well-aligned for most tasks considered). I think we’re largely on the same page here as per the “warm start” argument.
> > - There is no analysis of whether/how multi-modality arises and how BDFormer leverages this -- the following is a reasonable narrative, but it is not substantiated by any analysis as far as I can tell, so it reads somewhat post-hoc:
> >   > In complex graphs, a single class often exhibits a multimodal distribution spanning multiple latent clusters.

---

> > > ### Author Response · Authors · 2026-04-06
> > >
> > > We deeply appreciate your valuable feedback. Below is our response to your concerns.
> > >
> > > We are glad to be on the same page regarding the "warm start. We also completely agree that the 'multi-modality' claim requires explicit empirical substantiation to demonstrate how the model mathematically captures and leverages it.
> > >
> > > To address this, we provide the following clarifications based on our core architectural mechanism and empirical findings:
> > >
> > > **Mechanism Design**: To proactively capture multimodal intra-class clusters, BDFormer was deliberately designed to prevent anchors from trivially collapsing into a 1-to-1 label mapping.
> > > - Capacity and Symmetry Breaking (Eq. 8): When initializing $k$ latent anchors from $c$ class centroids (where $k > c$), the $c$ centroids are naturally duplicated. To proactively prevent these duplicates from remaining identical, we project them via an MLP and inject independent Gaussian noise $\epsilon$ (i.e., $B^{(0)} = MLP_{init}(C) + \epsilon$). This explicitly breaks the homogeneity among anchors originating from the same class.
> > > - Forcing Topological Clusters (Eq. 17): To ensure these perturbed anchors converge to meaningful structures rather than random noise, the Spectral Block Regularization is applied. Minimizing the $g$ smallest eigenvalues mathematically forces the anchors to form exactly $g$ distinct connected components (semantic blocks) in the latent space.
> > >
> > > **Empirical Substantiation**: If BDFormer were merely performing tautological label fitting, the optimal number of latent anchors ($k$) and semantic blocks ($g$) would simply equal the target class count ($c$). However, our grid search results explicitly demonstrate a data-driven decoupling:
> > > - Sufficient Micro-Capacity ($k \gg c$): Across all datasets, the optimal number of latent anchors ($k$) consistently vastly exceeds the number of classes ($c$). **This mathematical capacity guarantees that a single nominal class is allowed to exhibit a multimodal distribution spanning multiple latent anchors (clusters)**, inherently accommodating the diverse intra-class variance you correctly pointed out.
> > > - Adaptive Macro-Block Formations ($g$ vs. $c$): Governed by the Spectral Regularization, the optimal number of blocks ($g$) adapts dynamically to the intrinsic topology rather than human-defined labels. This results in dataset-specific behaviors that align with the objective characteristics of these graphs:
> > >   + When topological fragmentation is severe (e.g., Chameleon, PubMed): For highly heterophilic graphs like Chameleon, nodes of the same class are inherently scattered across diverse topological neighborhoods. Similarly, for datasets with coarse labels like PubMed ($c=3$), a single label functionally encompasses a massive, topologically diverse sub-network. To accommodate these structurally diverse, multimodal sub-communities within the same nominal classes, the model aggressively creates additional blocks ($g > c$).
> > >   + When fine-grained labels share macro-topological similarities (e.g., ogbn-Arxiv): Conversely, on ogbn-Arxiv ($c=40$), the fine-grained CS sub-fields heavily cross-cite and naturally form fewer major macro-communities in the topological space. The model adaptively extracts fewer macro-topological blocks ($g < c$), leveraging our dual-channel architecture by leaving the fine-grained micro-discrimination to the local GNN channel.
> > >   + When macro-motifs broadly align with class sizes (e.g., Squirrel, Paris): Even when $g \approx c$, the vast underlying anchor capacity ($k \gg c$) ensures that these blocks represent complex topological aggregations rather than a trivial 1-to-1 label assignment.
> > >
> > > In short, this adaptivity and mechanism design substantiate our claim: BDFormer is designed to learn and empirically leverages the intrinsic multimodal topological contexts. We will incorporate this specific analysis—connecting Eq. 8, Eq. 17, and the empirical adaptability of $(k, g)$ vs. $c$—into the revision.

---

### Decision · Program_Chairs · 2026-04-30

**Decision:**

Accept (regular)

**Comment:**

All four reviewers converged on positive scores after a thorough rebuttal, recognizing the clear motivation, novel block-diagonal structure, and comprehensive empirical validation. This paper makes a nice contribution to the Graph Transformer literature by providing a principled structural lens on representation propagation and an efficient, scalable architecture that consistently delivers strong results, and I recommend acceptance.